# Differentially Private Federated Clustering with Random Rebalancing

## Abstract

Federated clustering aims to group similar clients into clusters and produce one model for each cluster. Such a personalization approach typically improves model performance compared with training a single model to serve all clients, but can be more vulnerable to privacy leakage. Directly applying client-level differentially private (DP) mechanisms to federated clustering could degrade the utilities significantly. We identify that such deficiencies are mainly due to the difficulties of averaging privacy noise within each cluster (following standard privacy mechanisms), as the number of clients assigned to the same clusters is uncontrolled. To this end, we propose a simple and effective technique, named `RR-Cluster`, that can be viewed as a light-weight add-on to many federated clustering algorithms. `RR-Cluster` achieves reduced privacy noise via _randomly rebalancing_ cluster assignments, guaranteeing a minimum number of clients assigned to each cluster. We analyze the tradeoffs between decreased privacy noise variance and potentially increased bias from incorrect assignments and provide convergence bounds for `RR-Cluster`. Empirically, we demonstrate that `RR-Cluster` plugged into existing federated clustering algorithms results in significantly improved privacy/utility tradeoffs across both synthetic and real-world datasets.

## 1 Introduction

Clustering is a critical approach for adapting to varying client distributions in federated learning (FL) by learning one model for each group of similar clients (Li et al., 2020a; Kairouz et al., 2021). It usually outperforms the paradigm of learning a single global model, as it outputs several models to serve heterogeneous client population (Ghosh et al., 2020). Federated clustering typically works by identifying clusters the clients belong to and incorporating model updates into the corresponding cluster models, iteratively. However, performing clustering requires clients to transmit additional data-dependent information, making the learning system more vulnerable to privacy leakage.

In this work, we aim to finally output $k$ cluster models such that the set of these $k$ models satisfies global client-level differential privacy (DP) (Dwork et al., 2006; McMahan et al., 2018) (with a trusted central server). Despite recent attempts privatizing federated clustering to guarantee DP (Li et al., 2023; Augello et al., 2023), we identify that a fundamental tension between DP and clustering still remains—DP tries to hide individual client's contributions to the output models, whereas clustering may expose more client information (i.e., client model updates or cluster indices), especially if the cluster size is small. This could result in much larger effective noise when privatizing those client updates, which can degrade the overall privacy/utility tradeoffs. For instance, we observe significant utility drops if directly applying standard DP mechanisms on top of existing federated clustering algorithms (Figure 1 and Section 5).

Motivated by the above observation, we propose a simple and light-weight technique, `RR-Cluster`, that avoids too few clients being assigned to any cluster via randomly sampling a subset of client model updates belonging to large clusters into small ones. It guarantees that each cluster has at least $B$ assigned model updates at each round. Hence, the effective privacy noise after averaging the updates within each group would become smaller. Such data-independent random rebalancing step can be applied on top of various clustering algorithms where the server aggregates model updates within each cluster based on learnt cluster assignments, without extra privacy or communication costs. We note that a key hyperparameter here is

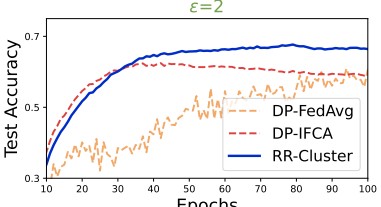 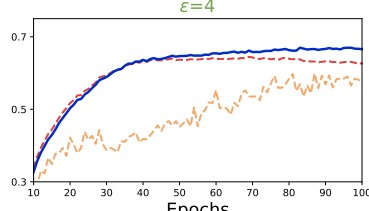 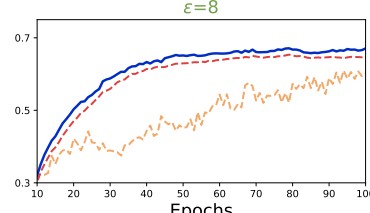

Figure 1: Test accuracy on FashionMNIST (Xiao et al., 2017) with a neural network model. Directly privatizing previous clustering method (Ghosh et al., 2020) (DP-IFCA) degrades model performance significantly, even underperforming the non-personalized approach (DP-FedAvg (McMahan et al., 2018)) for some privacy budgets. The proposed `RR-Cluster` plugged into IFCA achieves higher accuracy in private settings under various $\varepsilon$'s.

$B$, the lower bound of the number of clients assigned to each cluster at each round. When $B$ increases, we reduce privacy noise at the cost of potentially increasing clustering bias by potentially moving clients from correct clusters to incorrect ones. Theoretically, we analyze the effects of $B$ in the convergence bounds when plugging `RR-Cluster` into a strong federated clustering algorithm (Ghosh et al., 2020). Empirically, we find that (1) small values of $B$ can improve privacy/utility tradeoffs significantly, (2) and that the potential clustering bias from `RR-Cluster` does not hurt the learning process too much, especially in cases where the original assignment before rebalancing may not be accurate.

Our contributions are summarized as follows. We propose `RR-Cluster`, a simple technique to improve privacy/utility tradeoffs for federated clustering by randomly sampling model updates from large clusters to small ones (Section 3). It can be used as a light-weight add-on on top of various federated clustering algorithms. We theoretically analyze the privacy and convergence bounds of `RR-Cluster` in convex settings, with noise calibrated to the joint sensitivity of all $k$ released models (Section 4.1), revealing a tradeoff between reduced privacy noise and increased bias (Section 4). Empirically, we demonstrate the effectiveness of RR-Cluster across seven datasets spanning image classification (FashionMNIST, EMNIST, CIFAR-10, CIFAR-100, TinyImageNet), text (Shakespeare), and synthetic regression, under both natural and controlled heterogeneous data distributions (Section 5). Across all benchmarks, `RR-Cluster` outperforms vanilla private clustering baselines by a large margin.

## 2    Related Works and Background

**Federated Clustering and Personalization.**    Federated clustering is a critical approach for federated networks by learning one model for each group of similar clients, which can benefit various downstream tasks including personalization (Dennis et al., 2021). The IFCA method (Ghosh et al., 2020) is one of the most popular clustering algorithm, where each client 1) computes the losses of all cluster models on the local dataset and 2) selects the cluster having model of minimum test loss as their belonged cluster. Other clustering approaches leverage different distance measures for the cluster assignment (Long et al., 2023) or incorporate an additional global model to improve performance (Ma et al., 2023). While these clustering methods significantly enhance the performance of federated systems, challenges remain to guarantee differential privacy (DP) for the entire learning pipeline. In Section 5, we show how plugging `RR-Cluster` into some of these clustering methods can achieve better test performance relative to directly privatizing these clustering methods.

**Differentially Private Distributed/Federated Clustering.**    It is critical to ensure (differential) privacy when clustering distributed and sensitive data, which has been studied extensively in prior works mostly for data analytics (e.g., Nissim & Stemmer, 2018; Stemmer, 2021; Huang & Liu, 2018; Balcan et al., 2017; Chang et al., 2021; Cohen-Addad et al., 2022). However, joint clustering and *learning* algorithms under privacy constraints are generally less explored. In the context of federated learning, existing works have considered private federated clustering in different settings. For instance, the DCFL algorithm (Augello et al., 2023) combines a dynamic clustering method with DP under global DP. Malekmohammadi et al. (2025) propose DPCFL, which uses full-batch local training in the first round and a GMM-based soft clustering

of client updates to mitigate the impact of DP noise. However, DPCFL focuses on the *sample-level DP* setting, whereas our work targets *client-level global DP* and addresses a different set of privacy and clustering challenges. We note that some works may provide insufficient privacy protection without privatizing all the raw-data-dependent intermediate information in the clustering process (e.g., cluster assignment identifiers) (Li et al., 2023; Luo et al., 2024). Despite the existence of private federated clustering algorithms, issues related to high sensitivity of averaged model updates that belong to small clusters (thus resulting in large privacy noise) still remain. Some works target at sample-level DP (Liu et al., 2022) or local DP (He et al., 2023), different from our setting focusing on client-level global DP with a trusted central server, as defined below.

**Recent Clustered FL and Adaptive DP Methods.** Recent works improve cluster *discovery* via gradient-based partitioning (Kim et al., 2024), integrated CFL pipelines (Guo et al., 2025), or shift-aware descriptors (Fenoglio et al., 2025), while adaptive clipping (Andrew et al., 2021) and time-adaptive privacy spending (Kiani et al., 2025) optimize the clipping or budget schedule; cross-silo personalization (Liu et al., 2022) assumes a different (silo-level) trust model. None of them lower-bounds the number of private updates each cluster aggregates, which is the bottleneck we target. We compare with client-level DP adaptations of Malekmohammadi et al. (2025), Kim et al. (2024), and Fenoglio et al. (2025) in Section 5.2.

**Definition 1** (Differential Privacy (DP) (Dwork et al., 2006)). *A randomized algorithm $\mathcal{M} : U \to \mathbb{R}^{k \times d}$ is $(\varepsilon, \delta)$-DP if for all adjacent datasets $D, D' \in U$ and the algorithm output $O \in \mathbb{R}^{k \times d}$,*

$$\Pr(\mathcal{M}(D) \in O) \leq e^{\varepsilon} \cdot \Pr(\mathcal{M}(D') \in O) + \delta.$$

For client-level global DP, we define adjacent dataset $D'$ and $D$ by adding or removing any single client, which is a commonly-used privacy notion in federated settings (McMahan et al., 2018). In the clustering setting, we consider the output $\mathcal{M}(D) \in \mathbb{R}^{k \times d}$ to be $k$ clustered models, each in $\mathbb{R}^d$.

Additionally, we adopt Rényi differential privacy (RDP) (Mironov, 2017) for privacy accounting. For completeness, we include its formal definition below, and restate the existing RDP-to-DP conversion theorem in Theorem 3 in the appendix.

**Definition 2.** *($(\lambda, \epsilon)$-Rényi Differential Privacy (Mironov, 2017)) A randomized mechanism $\mathcal{M} : U \to \mathbb{R}^{k \times d}$ is $(\lambda, \epsilon)$-RDP if for all adjacent datasets $D, D' \in U$ and the algorithm output $O \in \mathbb{R}^{k \times d}$,*

$$D_\lambda(\mathcal{M}(D)\|\mathcal{M}(D')) = \frac{1}{\lambda - 1} \log \mathbb{E}_{o \sim \mathcal{M}(D)} \left[ \left( \frac{\Pr(\mathcal{M}(D) = o)}{\Pr(\mathcal{M}(D') = o)} \right)^{\lambda - 1} \right] \leq \epsilon.$$

**Privacy Setting and Threat Model.** In this work, we follow the standard and widely-adopted setting of client-level differential privacy with a trusted central server (McMahan et al., 2018; Kairouz et al., 2021). Our goal is to protect client data from an external adversary, who can observe the final output of the entire federated learning process, which in our case is the set of $k$ final cluster models $\{\hat{\theta}_j^T\}_{j \in [k]}$. We note that this setting is different from local DP, where the server is untrusted and clients privatize their updates before transmitting information. In our threat model, the central server is trusted and is not considered an adversary. We treat any external observer as the adversary, who only sees the final cluster models and attempts to infer whether any particular client participated in the federated training.

# 3 Private Federated Clustering with Random Rebalancing

In this section, we first introduce the general framework of `RR-Cluster` (Section 3.2) and then present an instantiation of `RR-Cluster` by plugging it into a popular clustering method IFCA (Ghosh et al., 2020) (Section 3.3). Finally, we discuss tradeoffs introduced by `RR-Cluster` around privacy noise reduction and clustering bias (Section 3.4).

## 3.1 Problem Formulation

In this work, we study the common federated clustering objective, i.e., grouping $M$ clients into $k$ groups and outputting one model for each group. We assume there are $k$ underlying data distributions $\{\mathcal{D}^0, \cdots, \mathcal{D}^{k-1}\}$, and

---

**Algorithm 1:** The Proposed Method: `RR-Cluster`

---

**1** **Input:** total communication round $T$, number of clusters $k$, number of clients $M$, init clusters model $\{\theta_j^0\}_{j \in [k]}$, client sampling rate $q$, parameter clipping bound $C_\theta$, rebalancing threshold $B$ ($1 \le B \le \frac{qM}{k}$). The specific $s_i$ mentioned in Line 5 depends on the underlying clustering algorithm. See details of $s_i$ in Algorithm 2.

**2** **for** $t = 0, \cdots, T-1$ **do**

**3**     Server samples a subset of clients $M^t$ with probability $q$

**4**     Server sends $\{\theta_j^t\}_{j \in [k]}$ to sampled clients

**5**     Client $i \in M^t$ sends back model updates $\Delta \theta_i^t$ and other data-dependent information $s_i$

**6**     Server privatizes $\{s_i\}_{i \in M^t}$ into $\{\widetilde{s}_i^t\}_{i \in M^t}$ if $\{s_i\}_{i \in M^t}$ directly queries raw client data

**7**     Server partitions $M^t$ into $k$ groups (with client indices in $\{S_j^t\}_{j \in [k]}$) based on $\widetilde{s}_i$

**8**     Server determines large/small groups comparing $|S_j^t|$ with $B$, $j \in [k]$

**9**     Server samples clients from large groups to insert into small ones, i.e., rebalancing cluster assignments s.t. $|S_j^t| \ge B$ for all $j \in [k]$

**10**     Server clips client updates $\Delta \bar{\theta}_i^t \leftarrow \frac{\Delta \theta_i^t}{\max(1, \|s\|/C_\theta)}$, $i \in M^t$

**11**     Server aggregates client updates $\Delta \widetilde{\theta}_j^t \leftarrow \frac{1}{|S_j^t|} \left( \sum_{i \in S_j^t} \Delta \bar{\theta}_i^t + \mathcal{N}(0, (\sqrt{5} C_\theta)^2 \sigma_\theta^2) \right)$, $j \in [k]$

**12**     Server uses noisy updates to update cluster models $\theta_j^{t+1} \leftarrow \theta_j^t + \gamma \Delta \widetilde{\theta}_j^t$, $j \in [k]$

**13** **return** *cluster models* $\{\theta_j^T\}_{j \in [k]}$

---

the $M$ client indices are partitioned into $k$ disjoint sets $\{S_0, \cdots, S_{k-1}\}$ indicating which clusters (distributions) clients belong to. Local data $D_i$ of client $i \in [M]$ follow the distribution $\mathcal{D}^j$ if $i \in S_j$. Let $f(\theta; z)$ be the loss function with model parameter $\theta \in \mathbb{R}^d$ and data point $z$. For each client $i \in [M]$, we define their local empirical loss $F_i(\theta)$ as $F_i(\theta) := \frac{1}{|D_i|} \sum_{z \in D_i} f(\theta; z)$. The optimization objective of cluster $j$ is: $F^j(\theta) := \frac{1}{|S_j|} \sum_{i \in S_j} F_i(\theta), j \in [k]$. We search for optimal parameters $\theta_j^*$ for each cluster $j \in [k]$ as $\theta_j^* := \arg\min_\theta F^j(\theta)$. For federated clustering algorithms, the server typically maintains cluster models $\{\theta_j\}_{j \in [k]}$, which are periodically aggregated from corresponding client models based $\{S_j\}_{j \in [k]}$. A complete table of notations is provided in Appendix E.

### 3.2 `RR-Cluster` **Algorithm**

We introduce the differentially private federated clustering framework and the proposed `RR-Cluster` method. As `RR-Cluster` is compatible with many clustering methods, we present a general algorithm to illustrate the main idea in this section and present an instantiation with IFCA (Ghosh et al., 2020) in Section 3.3.

At a high level, federated clustering jointly updates $k$ cluster model parameters and cluster assignments of clients, as summarized in Algorithm 1. In the $t$-th round, server samples $qM$ clients, and sends the current $k$ cluster models to each of them. Each client $i$ identifies the cluster they belong to and optimizes that cluster model's parameters using local data. Client $i$ then send the model update $\Delta \theta_i^t$ to the server as well as other information necessary for clustering (e.g., their cluster ids) (Line 5). Without our proposed plug-in, the server would identify model updates corresponding to each cluster (Line 7 where the set of client indices that belong to cluster $j$ is denoted as $S_j^t$), privatize and aggregate those updates within each cluster (Line 10 to Line 12), and output new cluster models $\{\theta_j^{t+1}\}_{j \in [k]}$. This procedure directly privatizes existing federated clustering algorithms that jointly updates $\{S_j^t\}_{j \in [k]}$ and $\{\theta_j^t\}_{j \in [k]}$.

However, one fundamental challenge of these clustering algorithms is that the cardinality of the set of updates corresponding to each cluster (i.e., $|S_j^t|, j \in [k]$) is not controlled. Therefore, the privacy noise needed to hide client model updates within each cluster (i.e., effective noise added after averaging model updates) could be large. To address this issue, we propose a simple and effective plug-in, `RR-Cluster` (highlighted in red), that uniformly randomly samples model updates from large groups (i.e., $|S_j^t| > B$) and move them to small ones.

---

**Algorithm 2:** `RR-Cluster` (IFCA)

---

**1**    **Input:** The same as Algorithm 1

**2**    **for** $t = 0, \cdots, T-1$ **do**

**3**       Same steps as in Algorithm 1, Line 3 - Line 4

**4**       **for** *client $i \in M^t$ in parallel* **do**

**5**           Determines its cluster: $\hat{j} \leftarrow \underset{j \in [k]}{\arg\min} \, F(\theta_j^t, D_i)$ and embeds $\hat{j}$ into one-hot: $s_i^t \leftarrow \mathbf{1}_{\{j=\hat{j}\}}$

**6**           $\theta_i^t \leftarrow$ local training$(D_i, \theta_{\hat{j}}^t), \quad \Delta\theta_i^t \leftarrow \theta_i^t - \hat{\theta}_{\hat{j}}^t$

**7**           Send $\Delta\theta_i^t$ and $s_i^t$ back to server

**8**       Server privatizes $s_i$ as $\widetilde{s}_i^t \leftarrow \frac{s_i^t}{\max(1, \|s\|/C_s)} + \mathcal{N}(0, C_s^2 \sigma_s^2)$ for $i \in M^t$

**9**       Server updates cluster assignments: $S_j^t$.append$(i)$ where $j \leftarrow \underset{j \in [k]}{\arg\max} \, \widetilde{s}_i[j]$ for $i \in M^t$

**10**       Server determines large/small groups $j_l = \{j \, \big| \, |S_j^t| \geq B, j \in [k]\}; \; j_s = [k] \setminus j_l$

**11**       Server samples client indices assigned to large clusters $\bigcup_{j \in j_l}\{S_j^t\}$ uniformly at random to expand small ones $\{S_j^t\}_{j \in j_s}$ such that $|S_j^t| = B$ for $j \in j_s$

**12**       Server privatizes $\Delta\theta_i^t$ and update cluster model $\theta_j^t$ as in Algorithm 1, Line 10 - Line 12

**13**    **return** *cluster models* $\{\theta_j^T\}_{j \in [k]}$

---

With client sample rate $q$, we guarantee that each cluster has a minimum of $B$ client model updates (as long as the input $B$ satisfying $1 \leq B \leq qM/k$), thus reducing privacy noise by averaging across at least $B$ updates to update each $\theta_j^t$. Note that given $B \leq \frac{qM}{k}$, any client update can only be sampled and reassigned to another cluster at most once at each round. Given the modularity of `RR-Cluster`, it can preserve the communication-efficiency of any federated clustering method that it is plugged into. Furthermore, we discuss the tradeoffs introduced by the rebalancing threshold $B$ in Section 3.4.

We formalize the rebalancing step (Line 9) as a randomized set map $R_B$.

**Definition 3** (Rebalancing map $R_B$). *At round $t$, let $j_l = \{j : |S_j^t| \geq B\}$ and $j_s = [k] \setminus j_l$. $R_B$ forms a donor pool $P$ by taking, from each cluster $j \in j_l$, a uniformly random subset of exactly $(|S_j^t| - B)$ of its indices (its surplus), sampled without replacement; each cluster in $j_s$ draws from $P$ uniformly without replacement until it reaches size $B$. Donated indices are* moved *(removed from the donor, inserted into exactly one recipient), never duplicated; feasibility is guaranteed by $B \leq qM/k$.*

By construction, the output of $R_B$ is a partition of the sampled clients: since updates are moved rather than duplicated and donors give away only their surplus, each client update appears in exactly one final cluster aggregate, and each donor cluster retains at least $B$ updates. Both properties are used in the sensitivity analysis in Appendix A.2.

### 3.3   `RR-Cluster` **Plugged into IFCA**

Having established the high-level structure of `RR-Cluster`, we now plug the random rebalancing idea into one popular clustering method IFCA (Ghosh et al., 2020) as an example, presented in Algorithm 2. We also empirically evaluate the performance of `RR-Cluster` plugged into other federated clustering algorithms in Section 5. In IFCA, at each iteration, each client independently determines the cluster they belong to by selecting the model with the smallest loss evaluated using local data, as shown in Line 5. Each selected client $i$ additionally communicates their cluster identifier to the server organized as a one-hot vector $s_i \in \mathbb{R}^k$, which is privatized along with model parameters.

**Other Differentially Private Mechanisms.** There are various privatization algorithms for model parameters $\theta$ and cluster identifier $s_i$'s. `RR-Cluster` is agnostic of the specific choice of private mechanisms. For $\theta \in \mathbb{R}^d$, it is a common practice to use the Gaussian mechanism after clipping to upper bound the $L_2$ norm of parameters. However, for cluster identifiers $s_i^t$, which is a $k$-dimensional one-hot discrete vector, it is also reasonable to adopt the exponential mechanism. Though we use the Gaussian mechanism in experiments (Section 5), `RR-Cluster` achieves better performance regardless of how to privatize identifiers, as it reduces DP noise for model parameters.

### 3.4  Effects of $B$: Tradeoffs Between Privacy Noise and Clustering Bias

Our method mainly benefits from the reduction of privacy noise after inserting model updates sampled from large groups $\{S_j\}_{j \in j_l}$ into the small ones $\{S_j\}_{j \in j_s}$ (Line 11). However, this approach may introduce bias if the updates that are initially correctly assigned to large groups are incorrectly re-assigned to one of the small ones. We discussed this issue both theoretically (Section 4.2) and empirically through a case study on synthetic data (Section 5.3). Empirically, we observe that benefits of reduced privacy noise outweigh potential bias. We also note that the donor selection in $R_B$ is uniform over indices and data-independent, hence incurs no extra privacy cost (Section 4.1); data-dependent alternatives (e.g., loss- or gradient-based selection) must be additionally privatized, and we compare with them in Section 5.3.

Additionally, we note that the such a random rebalancing step may not introduce much error or bias, as the initial cluster assignment (before rebalancing) may be incorrect anyway, which is a common issue for (federated) clustering sometimes known as model collapse (Wu et al., 2022). That is, the clustering algorithm starts with $k$ models but only a subset of the $k$ models get effectively trained and the solutions get gradually biased towards those well-trained models. As empirically demonstrated in Table 6, a subset of models can collapse after training. As our method maintains a minimum number of updates assigned to each cluster, one side effect is that it ensures each cluster receives some updates and thus would not be under-trained even if the updates may be slightly biased. Though `RR-Cluster` is designed for private training, in Section 5, we show that our method remains competitive or even outperforms other baselines in non-private training as well, due to this side effect.

## 4  Theoretical Analysis

### 4.1  Privacy Analysis

We state the privacy guarantees for the proposed method Algorithm 1 in terms of user-level global DP and RDP. In each communication round, conditioned on a randomly sampled subset of clients, there are two other random processes: (1) privatizing cluster identifiers and (2) privatizing model parameters. Hence, the privacy loss per round is accumulated from the composition of these two processes and the client subsampling step. First, we have the following proposition.

**Proposition 1.** *At each round of `RR-Cluster`, given a subset of selected clients $M^t$ as input, a randomized mechanism $\mathcal{M}$ that outputs $k$ new clustered models satisfies $(\alpha, \varepsilon_1 + \varepsilon_2)$-RDP if: (1) we set the noise scale for model parameters to $\sigma_\theta = \sqrt{\frac{\alpha}{2\varepsilon_2}}$ and (2) we use any cluster identifier privatization method that satisfies $(\alpha, \varepsilon_1)$-RDP.*

The above result directly follows from adaptive composition properties of RDP (Mironov, 2017) combined with the sensitivity bound of the concatenated cluster updates (Appendix A.2). As discussed in Section 3.2, we can adopt existing randomized algorithms to privatize cluster identifiers, as long as it achieves $(\alpha, \varepsilon_1)$-RDP. Using these noisy cluster identifiers, we identify large and small clusters and perform random rebalancing. Given rebalanced clusters, we release the $k$ cluster updates through a single Gaussian mechanism on their concatenation. Since rebalancing makes the partition data-dependent, a client can influence up to two cluster aggregates. We therefore bound the joint $L_2$ sensitivity of all $k$ updates by $\sqrt{5}\,C_\theta$ and calibrate the noise in Algorithm 1 accordingly, which yields $(\alpha, \varepsilon_2)$-RDP for the joint release per round. See Lemma 2 in Appendix A.2 for the detailed sensitivity analysis. All experiments in Section 5 use this calibration.

To derive the final privacy guarantee of `RR-Cluster`, we analyze (1) the client sampling process and (2) the composition of $T$ communication rounds. Note that the randomized mechanism $\mathcal{M}$ in Proposition 1 takes as input the subset of clients $M^t$. Thus, given the input of total $M$ clients, we can exploit randomness during client selection and derive a stronger privacy guarantee. Consider $\mathcal{M}$ as a black-box mechanism, we can apply privacy amplification theorem of RDP under sampling without replacement (Wang et al., 2019). We then apply composition theorem (Mironov, 2017) to account for total $T$ communication rounds and derive the final RDP privacy guarantee as follows.

**Theorem 1.** *Suppose a randomized mechanism $\mathcal{M}$ that inputs a subset of selected clients $M^t$ and outputs $k$ new clustered models satisfies $(\alpha, \varepsilon_1(\alpha) + \varepsilon_2(\alpha))$-RDP for a single communication round of `RR-Cluster`. Given*

*a client sampling ratio $q$ and total $T$ communication rounds, `RR-Cluster` algorithm satisfies $(\alpha, \varepsilon(\alpha))$-RDP where*

$$\varepsilon(\alpha) \leq \frac{T}{\alpha - 1} \cdot \log\left(1 + q^2\binom{\alpha}{2}\min\Big\{4(e^{\varepsilon_1(2)+\varepsilon_2(2)} - 1), \quad e^{\varepsilon_1(2)+\varepsilon_2(2)}\min\{2, (e^{\varepsilon_1(\infty)+\varepsilon_2(\infty)} - 1)^2\}\Big\}\right.$$
$$\left. + \sum_{j=3}^{\alpha} q^j\binom{\alpha}{j}e^{(j-1)(\varepsilon_1(j)+\varepsilon_2(j))})\min\{2, (e^{\varepsilon_1(\infty)+\varepsilon_2(\infty)} - 1)^j\}\right).$$

Here, $\varepsilon$ is denoted as $\varepsilon(\alpha)$, as $\varepsilon$ is functionally determined by the choice of $\alpha$. We note that the above complex bound directly follows from Wang et al. (2019), which provides a precise non-asymptotic bound rather than relying on an asymptotic bound. In Appendix A.3, we provide a detailed proof of Theorem 1 and the privacy guarantee in terms of DP using RDP-DP conversion (Mironov, 2017).

### 4.2 Convergence of `RR-Cluster`

In this section, we provide convergence guarantees for our proposed Algorithm 2 (`RR-Cluster` plugged into IFCA (Ghosh et al., 2020)) and theoretically explore the tradeoffs between increased clustering bias and reduced privacy noise as a result of random rebalancing with a parameter $B$.

For simplicity, our convergence analysis assumes all clients participate in every round. Our theoretical guarantees rely on several standard assumptions (as detailed in Appendix B). We assume the loss functions are $L$-smooth and $\lambda$-strongly convex. We also assume the variance of loss functions, variance of gradients, and variance of cluster sizes are bounded by $\eta^2$, and $v^2$, and $\mu^2$, separately. For the differential privacy analysis, the clipping threshold $C_\theta$ is assumed to be larger than norm of the natural gradient, meaning that the clipping operation does not affect the model update. Additionally, `RR-Cluster` enforces $1 \leq B \leq qM/k$ by construction, and all subsequent analysis is under this condition. Furthermore, we assume the underlying clusters are well-separated by a margin $\Delta$ and that the initial models lie within a constant fraction of $\Delta$ from the optima (Assumption 6 in the appendix). The latter is a bounded-initialization (basin) condition rather than a near-optimality requirement, standard in analyses of IFCA (Ghosh et al., 2020), EM (Dempster et al., 1977; Balakrishnan et al., 2017), and Lloyd's $k$-means (Lloyd, 1982; Lu & Zhou, 2016).

**Lemma 1.** *Let client $i$ belong to $S_j^t$ by ground-truth at the $t$-th communication round ($t \in [T]$). Assume that we obtain the cluster model $\theta_j^t$ such that there exist $0 < \beta < \frac{1}{2}$ and $\Delta := \min_{j\neq j'}\|\theta_j^* - \theta_{j'}^*\|$ that satisfy $\|\theta_j^t - \theta_j^*\| < (\frac{1}{2} - \beta)\sqrt{\frac{\lambda}{L}}\Delta$. Suppose we add zero-mean Gaussian noise with variance $\sigma_s^2$ to cluster identifiers with clipping bound 1. Then at the $t$-th round, the probability of misclassifying client $i$'s update into any other cluster $j' \neq j$ is upper bounded by*

$$\tau = \frac{8\eta^2 k}{\beta^2\lambda^2\Delta^4} + \frac{\sigma_s k}{\sqrt{\pi}}\exp(-1/4\sigma_s^2) + \frac{k^2\mu^2}{(M/k - B)^2}. \tag{1}$$

We note that in practice, the total number of clients is much larger than the number of clusters, i.e., $M/k \gg 1$. Hence $\tau$ can be much smaller than 1. In addition, $\tau$ decreases with the increase of the model separation parameter $\Delta$, which is expected. We prove Lemma 1 in Appendix B.2.

**Theorem 2.** *Assume in a certain iteration $t$ of Algorithm 2 run in the enforced-occupancy regime $1 \leq B \leq qM/k$, we obtain $\theta_j$ such that $\|\theta_j - \theta_j^*\| < (\frac{1}{2} - \beta)\sqrt{\frac{\lambda}{L}}\Delta$ for some $\beta \in (0, \frac{1}{2})$. Denote $\theta_j^+$ to be the parameter in next iteration. Then, for any $j \in [k]$, with probability at least $1 - \delta_c$, we have*

$$\|\theta_j^+ - \theta_j^*\| \leq \left(1 - \frac{\gamma L(B - 2\tau M/\delta_c)}{2\rho}\right)\|\theta_j - \theta_j^*\| + \epsilon,$$
$$\epsilon = \underbrace{\frac{4v}{\sqrt{\delta_c(B\delta_c - 4\tau M)}} + \frac{6\tau\gamma L\Delta M}{\delta_c B} + \frac{8\gamma vk\sqrt{\tau kM}}{B\delta_c\sqrt{\delta_c}}}_{Bias\ Term} + \underbrace{\frac{\sqrt{5}\gamma\sigma_\theta C_\theta}{B}\sqrt{d + 2\sqrt{d\ln(\frac{4}{\delta_c})} + 2\ln(\frac{4}{\delta_c})}}_{Variance\ Term}. \tag{2}$$

We prove Theorem 2 in Appendix B.3.

**Remark 1** (Recovering the base method). *When the base assignment already satisfies $|S_j^t| \geq B$ for all $j$, rebalancing is inactive and `RR-Cluster` coincides with the base method. Instantiating the bound at the base method's realized minimum occupancy $B_0 := \min_{j,t} |S_j^t|$ recovers its dependence, while `RR-Cluster` replaces the uncontrolled $B_0$ with the controlled floor $B$.*

**Remark 2** (Explicit range of valid $B$). *The contraction condition $B > 2\tau(B)M/\delta_c$ involves $\tau(B)$ itself. For $B \leq M/(2k)$, $k^2\mu^2/(M/k - B)^2 \leq 4k^4\mu^2/M^2$, so the condition is implied by the explicit bound $B > \frac{2M}{\delta_c}(\tau_0 + \frac{4k^4\mu^2}{M^2}) =: B_{\min}$, where $\tau_0$ collects the two $B$-independent terms of Eq. equation 1. Hence Theorem 2 holds on the explicit window $(B_{\min}, M/(2k)]$, non-empty for large $M$.*

We note that this bound gets worse as the privacy noise $\sigma_\theta$ and $\sigma_s$ (embedded in $\tau$) increase. Additionally, note $\tau \propto \mathcal{O}(\Delta^{-4}M^{-2})$, then the error term $\epsilon$ with $\tau$ can be small if we have enough clients (large $M$) and a good cluster separation initialization (large $\Delta$). Further, we provide the final convergence results in Corollary 1 in the appendix. We discuss the effects of $B$ in the next paragraph.

**Discussions on the Bias/Variance Tradeoff.** Our method aims to achieve better tradeoffs between clustering bias and privacy noise variance by random rebalancing parameterized by $B$. The error term $\epsilon$ in Theorem 2 clearly exposes this relationship. Specifically, we can decompose $\epsilon$ into two parts: a bias term due to incorrect clustering and a variance term due to DP noise.

- **Bias term due to incorrect clustering:** The first three terms of $\epsilon$ depend on the incorrect clustering probability $\tau$, which increases as the clustering error $\tau$ grows. Since Lemma 1 shows $\tau$ is proportional to $\frac{1}{(M/k-B)^2}$, a larger $B$ could increases this bias. This aligns with the intuition that a larger $B$ indicates sampling more client updates into potentially wrong clusters.
- **Variance due to privacy noise:** The last term in $\epsilon$, $\mathcal{O}(\sigma_\theta/B)$, is directly proportional to the DP noise scale $\sigma_\theta$ and inversely proportional to $B$. This term decreases as $B$ increases.

As a result, the overall error $\epsilon$ shows how the clustering error and privacy noise affect model convergence. We observe that as $B$ increases, we accept potentially more biased cluster assignments in exchange for a significantly smaller privacy noise perturbation. This allows us to strike a better tradeoff between bias and variance, which we empirically demonstrate in Section 5.

## 5 Experiments

In this section, we evaluate `RR-Cluster` across diverse datasets and scenarios. Section 5.2 presents main results on image and text tasks, showing that our approach outperforms state-of-the-art DP clustering baselines in privacy/utility tradeoffs and clustering quality, and serves as a general plug-in to various federated clustering algorithms; Section 5.3 provides diagnostic analysis. We mainly report the average classification accuracy across all clients, with maximum/minimum client accuracies in Appendix D.1.

### 5.1 Experimental Settings

**Datasets and Hyperparameters.** We evaluate our method on 7 datasets overall, including five image datasets of varying difficulty, one text dataset, and one synthetic dataset as follows. We use FashionMNIST (Xiao et al., 2017) that consists of 10-class 70,000 grayscale images of clothes, and EMNIST (Cohen et al., 2017), an extended dataset of the original MNIST, containing 814,255 images of handwritten characters with 47 categories covering both digits and letters. For the text data, we use the Shakespeare dataset (McMahan et al., 2017b; Shakespeare, 2014) on a next-character prediction task over the texts of Shakespeare's plays and writings. To control data separability and illustrate `RR-Cluster`'s behavior in non-private settings in more detail, we generate synthetic datasets for a linear regression task. For more challenging image benchmarks, we further include CIFAR10 and CIFAR100 (Krizhevsky et al., 2009), consisting of 60,000 32×32 color images across 10 and 100 classes respectively, as well as TinyImagenet (Deng et al., 2009), which contains 100,000 64×64 color images across 200 classes.

To simulate different data distributions in real-world scenarios, we explore both manual partitions with a clear clustering structure as in previous works and partitions without clear cluster priors from previous FL

benchmarks. For FashionMNIST, we followed the setting in IFCA (Ghosh et al., 2020), using image rotations to simulate the underlying ground truth clusters. We adopt the same rotation-based partition scheme for CIFAR10, CIFAR100, and TinyImagenet. We partition EMNIST based on a Dirichlet distribution over the labels (TFF), and partition the Shakespeare dataset based on speaking characters in the play (Caldas et al., 2018). In both image tasks, we have 1000 clients in total; for the Shakespeare dataset, we use 300 clients due to limitations of speaking roles. For all experiments, we use client sample rate $q = 0.1$, and tune $k \in \{2, 4\}$ based on validation data. Notably, these benchmarks include diverse heterogeneity types: The EMNIST dataset mainly simulates label distribution skew (i.e., $P(y)$ varies), while the Shakespeare dataset exhibits concept shift (i.e., $P(y|x)$ varies) as different roles naturally show distinct linguistic patterns. The rotated FashionMNIST task further reinforces this by introducing concept shift through geometric transformations.

**Baselines.** We compare our methods with several competitive FL and federated clustering methods under DP: FedAvg (McMahan et al., 2017a), FedProx (Li et al., 2020b), FedPer (Arivazhagan et al., 2019), IFCA (Ghosh et al., 2020), FeSEM (Long et al., 2023) and FedCAM (Ma et al., 2023). FedAvg is a classical FL baseline learning a global Model. IFCA, FeSEM and FedCAM are strong federated clustering approaches. We additionally compare with three recent clustered-FL methods adapted to our client-level DP protocol, DP-CFL, DP-GradPart, and DP-FLUX (see Section 5.2). While we focus on clustering (outputting $k$ models for $M$ clients where $k \ll M$) as opposed to broad federated personalization, we still compare with FedPer, a personalization method as a reference point. We directly privatize these methods using standard subsampled Gaussian mechanisms guarantee the same client-level global DP across all approaches. For clustering methods, we privatize cluster identifiers similarly as in `RR-Cluster`. We do not compare with other personalization works that output $M$ models for $M$ clients.

## 5.2 Comparison with Strong Baselines

We compare our method with other state-of-the-art methods on FashionMNIST, EMNIST, and Shakespeare, setting the same hyperparameters (e.g., number of clients, local learning rates and iterations, communication rounds) for all methods and tuning the clipping bound per task and method; see Appendix C for details. The main paper focuses on clustering-based baselines; full results with proximal- and personalization-based methods are in Appendix D.2.

**Plugging `RR-Cluster` to Base Clustering Methods.** In Table 1, we use FashionMNIST rotation to create underlying clusters of the clients, following the setup in previous works (Ghosh et al., 2020; Kim et al., 2024) . For "Balanced Clusters", we randomly divide the dataset into 1000 clients grouped evenly into four ground-truth clusters with rotations $\{0, 90, 180, 270\}$ degrees; for "Imbalanced Clusters", into three clusters at a ratio of 2:1:1 with rotations $\{0, 90, 180\}$ degrees. We plug `RR-Cluster` into existing federated clustering methods (IFCA (Ghosh et al., 2020), FeSEM (Long et al., 2023) and FedCAM (Ma et al., 2023)) to demonstrate its compatibility. In both settings with balanced or imbalanced clusters, we see that `RR-Cluster` results in more accurate final models compared with directly privatizing the base clustering algorithms without random rebalancing. We highlight improvement numbers for each setting in green; see convergence curves in Figure 2.

We further compare with three recent clustered-FL methods adapted to our client-level DP protocol (middle panel of Table 1): DP-CFL (GMM-based soft clustering of Malekmohammadi et al. (2025) on privatized updates), DP-GradPart (privatized gradient-based partitioning of Kim et al. (2024)), and DP-FLUX (descriptors of Fenoglio et al. (2025) privatized as our cluster identifiers). `RR-Cluster` outperforms all three, as they improve cluster *discovery* but do not lower-bound per-cluster occupancy, leaving small-cluster aggregates dominated by DP noise. All `RR-Cluster` results use noise calibrated to the joint sensitivity $\sqrt{5}\, C_\theta$ (Section 4.1); standard deviations are in Appendix D.4.

**Scaling to Harder Image Benchmarks.** We further evaluate `RR-Cluster` on CIFAR10, CIFAR100, and TinyImagenet under the "Balanced Clusters" setup. As shown in Table 2, `RR-Cluster` plugged into IFCA consistently outperforms all baselines across all three datasets and both privacy budgets, demonstrating that it scales to more complex image classification tasks. Beyond the CNN used here, we also evaluate larger

| Methods | Balanced Clusters | | | Imbalanced Clusters | | |
|---|---|---|---|---|---|---|
| | $\varepsilon = 2$ | $\varepsilon = 4$ | $\varepsilon = 8$ | $\varepsilon = 2$ | $\varepsilon = 4$ | $\varepsilon = 8$ |
| DP-FedAvg | 60.88 | 61.50 | 62.72 | 60.80 | 61.53 | 61.59 |
| DP-IFCA | 62.68 | 64.46 | 65.35 | 56.12 | 58.05 | 59.63 |
| DP-FeSEM | 63.16 | 64.68 | 64.76 | 58.55 | 59.82 | 60.10 |
| DP-FedCAM | 49.70 | 56.86 | 61.54 | 43.98 | 47.51 | 50.35 |
| DP-CFL | 61.62 | 62.73 | 65.66 | 58.51 | 60.54 | 61.67 |
| DP-GradPart | 63.39 | 64.88 | 65.17 | 60.68 | 61.23 | 62.11 |
| DP-FLUX | 61.06 | 61.40 | 63.52 | 58.78 | 59.41 | 60.95 |
| RR-Cluster (IFCA) | 64.92 ↑2.24 | 66.26 ↑1.80 | 67.48 ↑2.13 | 61.06 ↑4.94 | 61.64 ↑3.59 | 62.75 ↑3.12 |
| RR-Cluster (FeSEM) | 63.38 ↑0.22 | 64.78 ↑0.10 | 64.80 ↑0.04 | 58.64 ↑0.09 | 60.12 ↑0.30 | 60.65 ↑0.55 |
| RR-Cluster (FedCAM) | 51.23 ↑1.53 | 57.49 ↑0.63 | 63.73 ↑2.19 | 45.01 ↑1.03 | 47.89 ↑0.38 | 50.59 ↑0.24 |

Table 1: Comparison with baselines on FashionMNIST. We validate the effectiveness of our method under various DP budget $\varepsilon$'s. RR-Cluster plugged into different clustering methods (the last panel) outperforms the corresponding baselines that directly privatize the base clustering algorithms without considering uncontrolled cluster sizes. The middle panel reports three recent clustered-FL methods adapted to our client-level DP protocol (see Section 5.2).

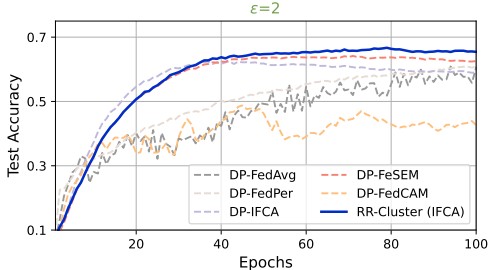 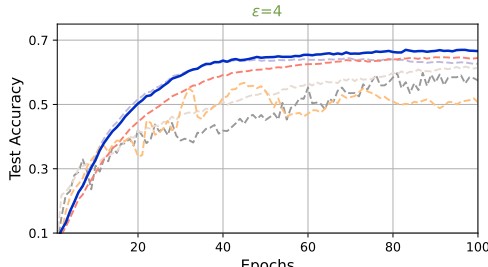

Figure 2: Convergence curves compared with strong baselines on FashionMNIST. We see that RR-Cluster achieves the better accuracies and faster convergence under both $\varepsilon$ values.

| Methods | CIFAR10 | | CIFAR100 | | TinyImagenet | |
|---|---|---|---|---|---|---|
| | $\varepsilon = 4$ | $\varepsilon = 8$ | $\varepsilon = 4$ | $\varepsilon = 8$ | $\varepsilon = 4$ | $\varepsilon = 8$ |
| DP-FedAvg | 52.89 | 54.48 | 14.05 | 14.49 | 13.67 | 14.06 |
| DP-IFCA | 53.91 | 56.77 | 16.29 | 17.47 | 15.79 | 16.76 |
| DP-FeSEM | 53.01 | 55.87 | 16.23 | 17.08 | 15.65 | 16.44 |
| DP-FedCAM | 52.50 | 54.73 | 16.36 | 17.04 | 15.71 | 16.38 |
| RR-Cluster (IFCA) | 56.45 ↑2.54 | 58.22 ↑1.45 | 17.20 ↑0.84 | 17.93 ↑0.46 | 16.73 ↑0.94 | 17.52 ↑0.76 |

Table 2: Comparison with baselines on CIFAR10, CIFAR100, and TinyImagenet datasets under various DP budgets $\varepsilon$. Across harder image tasks, RR-Cluster plugged into IFCA achieves the highest accuracy across all datasets and privacy levels, demonstrating that the random rebalancing technique scales well to harder tasks.

backbones, ResNet-18 and ViT-B initialized from public ImageNet pre-training and privately fine-tuned, on CIFAR10/100 in Appendix D.5; RR-Cluster outperforms the baselines on both architectures.

**Text Data with Ambiguous Cluster Patterns.** In Table 3, we study the performance on the Shakespeare dataset (Caldas et al., 2018), where each speaking role is a client. We do not control or manipulate the underlying clustering patterns a prior. Our method outperforms DP-FedAvg, DP-IFCA, and other clustering baselines that directly integrate DP on this task as well. Note that '−' indicates DP-FedCAM cannot converge, as it requires clients to

| Methods | $\varepsilon = 4$ | $\varepsilon = 8$ | $\varepsilon = 16$ |
|---|---|---|---|
| DP-FedAvg | 04.47 | 13.43 | 17.53 |
| DP-IFCA | 12.62 | 12.64 | 13.20 |
| DP-FeSEM | 10.97 | 12.99 | 15.89 |
| DP-FedCAM | - | - | 04.47 |
| RR-Cluster (IFCA) | 13.09 | 13.60 | 16.23 |

Table 3: On the Shakespeare dataset, our proposed approach achieves higher test accuracies compared with the baselines. '-' indicates failure to converge.

send both global and cluster model updates, resulting
in larger privacy noise; plugging `RR-Cluster` into FedCAM significantly reduces this noise (Table 1).

**Varying Heterogeneity.** Here, we create datasets with different degrees of heterogeneity. We partition the
EMNIST data following a Dirichlet distribution with parameters $\alpha = \{0.5, 0.1\}$ for 'Low / High Heterogeneity'
scenarios respectively. Table 4 shows that (1) most clustering methods outperform the global-model baseline
under higher heterogeneity, and (2) our method achieves the highest accuracies in almost all settings.

| Methods | Low Client Heterogeneity | | | High Client Heterogeneity | | |
|---|---|---|---|---|---|---|
| | $\varepsilon = 2$ | $\varepsilon = 4$ | $\varepsilon = 8$ | $\varepsilon = 2$ | $\varepsilon = 4$ | $\varepsilon = 8$ |
| DP-FedAvg | 26.47 | 30.95 | 32.61 | 23.82 | 24.46 | 25.54 |
| DP-IFCA | 62.23 | 65.39 | 65.93 | 32.53 | 35.60 | 36.19 |
| DP-FeSEM | 60.43 | 65.72 | 66.24 | 31.12 | 37.52 | 38.74 |
| DP-FedCAM | 57.35 | 57.22 | 61.26 | 23.66 | 24.55 | 29.70 |
| `RR-Cluster` (IFCA) | 64.96 ↑2.73 | 66.49 ↑0.77 | 66.87 ↑0.63 | 36.23 ↑3.70 | 37.69 ↑0.17 | 38.70 |

Table 4: Comparison with baselines on the EMNIST dataset. We see that `RR-Cluster` outperforms other approaches
especially when the privacy parameter is small (i.e., stronger privacy guarantees).

## 5.3 Diagnostic Analysis

**Hyperparameter Analysis.** As discussed before, the rebalancing hyperparameter $B$ introduces a tradeoff between privacy noise variance and clustering bias: a larger $B$ samples more updates into potentially wrong clusters, while a smaller $B$ leaves the contributions of clients in small clusters harder to hide. Choosing a proper value of $B$ $(1 \leq B \leq qM/k)$ is critical for our algorithm. We demonstrate the effects of $B$ on FashionMNIST (same rotation settings as Section 5.2) in Figure 3: there exists an optimal tradeoff point, and our method outperforms other private federated clustering baselines for a wide range of $B$'s.

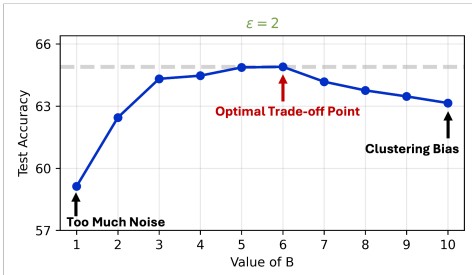

Figure 3: Ablation studies on the hyperparameter $B$ on the FashionMNIST dataset. There are a range of $B$'s that lead to improved performance. (Note that the baseline algorithm DP-IFCA in this setting achieves 62.68% accuracy.)

**A Practical Rule for Selecting $B$.** Solving the first-order condition of the two competing terms of Theorem 2 in $B$ (the $1/B$ noise decay and the $1/(qM/k - B)^2$ bias growth) yields the rule $B \leq \frac{qM}{3k}$. For FashionMNIST ($q = 0.1$, $M = 1000$, $k = 4$) this gives $B \leq 25/3$, matching the optimum $B = 6$ in Figure 3.

**Additional Diagnostics.** Appendix D.3 reports three further diagnostics: a donor-selection ablation (uniform vs. loss- or gradient-based selection), an oracle-assignment experiment isolating the pure rebalancing bias, and per-cluster accuracies under imbalance, where `RR-Cluster` raises minority-cluster accuracy by nearly 16 points at $\varepsilon = 2$.

**Guidance on Choosing the Privacy Budget $\varepsilon$.** For practitioners, $\varepsilon$ between roughly 1 and 10 is the commonly adopted range in DP practice (Abadi et al., 2016; McMahan et al., 2018; Ponomareva et al., 2023); operationally, one can fix $\delta \approx 1/M$ and read the smallest $\varepsilon$ meeting the target utility off Tables 1–4, and `RR-Cluster` widens this usable range by preserving utility at strict budgets.

**Clustering Accuracy.** In addition to the performance of downstream tasks (i.e, learnt model accuracies) reported in previous sections, we also compare different methods in terms of clustering accuracies—the portion of clients that are correctly clustered. We experiment on FashionMNIST with balanced rotation as we have access to the underlying clustering structures. We present the success rate in Table 5. Our method achieves the highest clustering accuracy with the increase of $B$. Note that our method can exhibit higher clustering accuracies than the baselines even in non-private cases. This implies the seemingly incorrect assignment caused by expanding small clusters may actually help to stabilize clustering, mitigating model collapse, which

| Methods | $\varepsilon = 0.5$ | $\varepsilon = 2$ | $\varepsilon = 4$ | $\varepsilon = 8$ | $\varepsilon = 16$ | $\varepsilon \to \infty$ |
|---------|------|------|------|------|-------|------|
| DP-IFCA | 34.37 | 42.18 | 39.06 | 40.62 | 42.18 | 75.00 |
| DP-FeSEM | - | 42.18 | 39.06 | 32.81 | 50.00 | 46.87 |
| RR-Cluster(IFCA) B=4 | 40.62 | 45.31 | 43.75 | 59.37 | 87.50 | 98.44 |
| RR-Cluster(IFCA) B=6 | 40.62 | 53.12 | 62.50 | 87.50 | 100.00 | 84.37 |
| RR-Cluster(IFCA) B=8 | **40.62** | **59.37** | **87.50** | **98.44** | **100.00** | **100.00** |

Table 5: Accuracy of clustering itself (i.e., percentage of correctly clustered clients) on FashionMNIST.

often occurs in clustering (Ma et al., 2023) (also discussed Section 3.4); the next paragraph illustrates this side effect on synthetic data.

**Case Study on Synthetic Data.** To demonstrate `RR-Cluster`'s side effect of mitigating model collapse, we conduct non-private experiments on synthetic linear regression data (generated as $\hat{y} = kx + b + \epsilon$), crossing balanced/imbalanced clusters with easy/hard separability. As shown in Table 6, IFCA suffers from model collapse even with $k=4$ specified: one of its cluster models remains untrained in both easy and hard settings, while `RR-Cluster` learns all four cluster models, and it still avoids collapse under imbalance at the cost of some additional bias. By guaranteeing each cluster at least $B$ updates, random rebalancing prevents the well-known under-training of cluster models even in non-private pipelines.

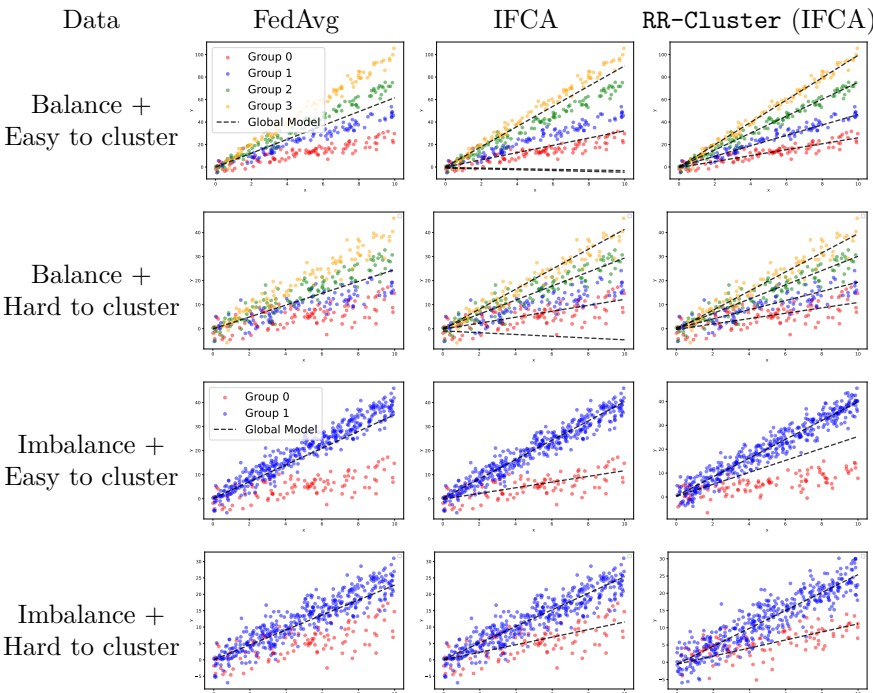

Table 6: Synthetic case study in non-private settings: datapoints are colored by ground-truth cluster and learned models are shown as black dashed lines. `RR-Cluster` learns all clusters while IFCA suffers from model collapse.

# 6 Conclusion, Limitations, and Future Work

In this paper, we introduce `RR-Cluster`, a simple add-on that achieves better privacy/utility tradeoffs for federated clustering via a random sampling mechanism that controls client contributions across clusters, thereby reducing the effective DP noise. We have demonstrated the effectiveness of `RR-Cluster` both theoretically and empirically across various tasks.

**Limitations and future work.** Our guarantee targets client-level global DP with a trusted server, and our convergence analysis covers strongly convex objectives; extending random rebalancing to other privacy units (e.g., sample-level or local DP) and to non-convex models is left for future work.

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

# A  Privacy Analysis

## A.1  RDP to DP Conversion

We restate existing RDP-to-DP conversion theorem, as follows.

**Theorem 3.** *(From RDP to $(\epsilon, \delta)$-DP (Mironov, 2017)) If mechanism $\mathcal{M}$ satisfies $(\lambda, \epsilon)$-RDP, then it also satisfies $(\epsilon + \frac{\log(1/\delta)}{\lambda - 1}, \delta)$-DP for any $\delta \in (0, 1)$.*

## A.2  Sensitivity Analysis

Continuing from Section 4.1, we study the sensitivity of our clustering updates each iteration for client-level DP. In global round $t$, we select $qM$ clients to update $k$ cluster models. Denote $\Delta\theta_i^t$ as the model updates from the $i$-th client at round $t$. By clipping, the $L_2$-norm of each update $\Delta\theta_i^t$ are limited within $[0, C_\theta]$. According to Algorithm 1, we add noise to the sum of cluster updates within each cluster. In the following, we bound the $L_2$ sensitivity of the *concatenated* vector of the $k$ per-cluster sums under the rebalancing map $R_B$ (Definition 3): since rebalancing makes the partition data-dependent, removing one client can alter more than one cluster, so we analyze the joint release directly instead of applying parallel composition cluster-wise.

**Lemma 2** (Joint sensitivity under $R_B$). *Let $G(\Theta) := \left( \sum_{i \in S_1^t} \Delta\bar{\theta}_i^t, \ldots, \sum_{i \in S_k^t} \Delta\bar{\theta}_i^t \right) \in \mathbb{R}^{k \times d}$ denote the concatenation of the $k$ per-cluster sums of clipped updates after rebalancing. Then, under client-level adjacency, the $L_2$ sensitivity of $G$ satisfies*

$$\max_{\Theta, \Theta'} \|G(\Theta) - G(\Theta')\|_2 \leq \sqrt{5}\, C_\theta.$$

*Proof.* Suppose that we have two adjacent client sets, $\Theta$ and $\Theta'$, where $\Theta'$ has one more/less client than $\Theta^t$. We denote the added/removed update as $\Delta\theta_A$. We couple the randomness of $R_B$ between $\Theta$ and $\Theta'$ such that all donor choices not involving $\Delta\theta_A$ coincide; by the disjointness of $R_B$ (Section 3.2), each update then appears in exactly one final aggregate, and at most one donor choice differs between the two runs. We note that adding/removing a client may or may not affect other groups of model updates based on our algorithm. Without loss of generality, we assume the added/removed client update belongs to cluster 1. We discuss two cases as follows.

**Case 1**: Adding or removing one client belonging to the groups assigned to cluster 1 does not trigger random sampling and rebalancing. For instance, cluster 1 after removing one client still satisfies that the cardinality is at least $B$. For the sensitivity of the function outputting $k$ clusters' model updates, we only need to consider cluster 1, as all other groups remain unchanged. Denote the set of model updates assigned to cluster 1 before removing $\Delta\theta_A$ as $\{\Delta\theta_{1,1}, \cdots, \Delta\theta_{1,n_1}, \Delta\theta_A\}$. Then the sensitivity of summing up model updates in cluster 1 can be upper bounded by

$$\max_{\Theta, \Theta'} \ \|(\Delta\theta_{1,1} + ... + \Delta\theta_{1,n_1} + \Delta\theta_A) - (\Delta\theta_{1,1} + ... + \Delta\theta_{1,n_1})\| = \|\Delta\theta_A\| \leq C_\theta. \tag{3}$$

If we add $\Delta\theta_A$, the same sensitivity holds. In this case only one coordinate of $G$ changes, so $\|G(\Theta) - G(\Theta')\|_2 \leq C_\theta$.

**Case 2**: The added/removed client triggers random sampling from large clusters to expand small ones. For example, removing a client results in cluster 1 originally having $B$ updates to only have $B - 1$ updates, in which case we would need to sample one model update from another cluster (denoted as $\Delta\theta'$) and insert it into cluster 1. The sensitivity of summing up model updates in cluster 1 is

$$\max_{\Theta, \Theta'} \ \|(\Delta\theta_{1,1} + ... + \Delta\theta_{1,n_1} + \Delta\theta') - (\Delta\theta_{1,1} + ... + \Delta\theta_{1,n_1} + \Delta\theta_A)\|$$
$$\leq \|\Delta\theta_A\| + \|\Delta\theta'\| \leq 2C_\theta. \tag{4}$$

Similarly, we have that the sensitivity corresponding to the cluster that we sample $\Delta\theta'$ from is $C_\theta$.

In this case, exactly two coordinates of $G$ change: cluster 1 by at most $2C_\theta$ and the donor cluster by at most $C_\theta$, while all remaining clusters are unchanged under the coupling. Hence

$$\|G(\Theta) - G(\Theta')\|_2 \leq \sqrt{(2C_\theta)^2 + C_\theta^2} = \sqrt{5}\,C_\theta. \tag{5}$$

Taking the maximum over both cases proves the lemma; each client contributes to at most two of the $k$ released aggregates, mirroring bounded-contribution analyses (Dwork et al., 2014; Wilson et al., 2020). Accordingly, Algorithm 1 calibrates the Gaussian noise to $(\sqrt{5}C_\theta)^2\sigma_\theta^2$, which yields Proposition 1 for the joint release of all $k$ cluster updates. $\qquad\square$

### A.3   Privacy Guarantees

Here we provide the proof of Theorem 1. We first introduce the definition of Rényi differential privacy and several theorems used in our privacy analysis.

**Definition 4** (Rényi differential privacy (Mironov, 2017))**.** *A randomized mechanism $f : \mathcal{D} \to \mathcal{R}$ is said to have $\varepsilon$-Rényi differential privacy of order $\alpha$, or$(\alpha, \varepsilon)$-RDP for short, if for any adjacent $D, D' \in \mathcal{D}$ it holds that*

$$D_\alpha(f(D)\|f(D')) = \frac{1}{\alpha - 1} \log \mathbb{E}_{x \sim f(D')} \left( \frac{f(D)}{f(D')} \right)^\alpha \leq \varepsilon. \tag{6}$$

**Theorem 4** (RDP adaptive composition (Mironov, 2017))**.** *Let $f : \mathcal{D} \to \mathcal{R}_1$ be $(\alpha, \varepsilon_1)$-RDP and $g : \mathcal{R}_1 \times \mathcal{D} \to \mathcal{R}_2$ be $(\alpha, \varepsilon_2)$-RDP, then the mechanism defined as $(X, Y)$, where $X \sim f(D)$ and $Y \sim g(X, D)$, satisfies $(\alpha, \varepsilon_1 + \varepsilon_2)$-RDP.*

**Theorem 5** (RDP with Gaussian mechanism (Mironov, 2017))**.** *If $f$ has sensitivity $C$, then the Gaussian mechanism $G_\sigma f(D) = f(D) + N(0, C^2\sigma^2)$ satisfies $\left(\alpha, \frac{\alpha}{2\sigma^2}\right)$-RDP.*

**Theorem 6** (RDP with subsampling (Wang et al., 2019))**.** *Given a randomized mechanism $\mathcal{M}$, and let the randomized algorithm $\mathcal{M} \circ subsample$ be defined as: (1) subsample: subsample with subsampling rate $q$; (2) apply $\mathcal{M}$: a randomized algorithm taking the subsampled dataset as the input. For all integers $\alpha \geq 2$, if $\mathcal{M}$ is $(\alpha, \varepsilon)$-RDP, then $\mathcal{M} \circ subsample$ is $(\alpha, \varepsilon_q)$-RDP where*

$$\begin{aligned}
\varepsilon_q \leq &\frac{1}{\alpha - 1} \log \left( 1 + q^2 \binom{\alpha}{2} \min \left\{ 4(e^{\varepsilon(2)} - 1), e^{\varepsilon(2)} \min \left[ 2, (e^{\varepsilon(\infty)} - 1)^2 \right] \right\} \right) \\
&+ \sum_{j=3}^{\alpha} q^j \binom{\alpha}{j} e^{(j-1)\varepsilon(j)} \min \left\{ 2, (e^{\varepsilon(\infty)} - 1)^j \right\}.
\end{aligned} \tag{7}$$

In Theorem 6, DP parameter $\varepsilon$ is denoted as $\varepsilon(\alpha)$, since $\varepsilon$ can be viewed as a function of RDP parameter $\alpha$ for $1 \leq \alpha \leq \infty$ (Wang et al., 2019).

**Theorem 7** (Converting RDP into $(\varepsilon, \delta)$-DP (Mironov, 2017))**.** *If $f$ is an $(\alpha, \varepsilon)$-RDP mechanism, it also satisfies $(\varepsilon + \frac{\log 1/\delta}{\alpha - 1}, \delta)$-differential privacy for any $0 < \delta < 1$.*

We now provide the overall privacy guarantee of `RR-Cluster`, stated in Theorem 1. We analyze the privacy budget by considering three steps: (1) adaptive composition of private identifier and private model parameters in a single round, (2) privacy amplification with subsampling ratio $q$, and (3) composition over $T$ rounds.

We first provide the privacy guarantee for a single round `RR-Cluster` without subsampling. We restate the Proposition 1 in the main text and provide the proof.

*Proof.* According to Theorem 5 applied to the concatenated vector of the $k$ cluster updates, with sensitivity $\sqrt{5}C_\theta$ (Lemma 2) and noise variance $(\sqrt{5}C_\theta)^2\sigma_\theta^2$, if using noise $\sigma_\theta = \sqrt{\frac{\alpha}{2\varepsilon_2}}$, we can calculate the privacy budget $\varepsilon$ as $\frac{\alpha}{2\sigma_\theta^2} = \varepsilon_2$. Which means the joint release of all $k$ new cluster models gives $(\alpha, \varepsilon_2)$-RDP. We then use adaptive RDP composition using Theorem 4 to compose budget $\varepsilon_1$ for identifiers and $\varepsilon_2$ for parameters to get $(\alpha, \varepsilon_1 + \varepsilon_2)$-RDP in total. $\qquad\square$

We then amplify the $(\alpha, \varepsilon_1 + \varepsilon_2)$-RDP budget with client subsampling with ratio $q$ and then compose it over $T$ rounds to provide overall privacy guarantee as follows. We denote $\varepsilon$ as $\varepsilon(\alpha)$ as in Theorem 6.

**Theorem 1.** *Suppose a randomized mechanism $\mathcal{M}$ that inputs a subset of selected clients $M^t$ and outputs $k$ new clustered models satisfies $(\alpha, \varepsilon_1(\alpha) + \varepsilon_2(\alpha))$-RDP for a single communication round of* `RR-Cluster`. *Given a client sampling ratio $q$ and total $T$ communication rounds,* `RR-Cluster` *algorithm satisfies $(\alpha, \varepsilon(\alpha))$-RDP where*

$$\varepsilon(\alpha) \leq \frac{T}{\alpha - 1} \cdot \log\left(1 + q^2 \binom{\alpha}{2} \min\left\{4(e^{\varepsilon_1(2)+\varepsilon_2(2)} - 1), \quad e^{\varepsilon_1(2)+\varepsilon_2(2)} \min\{2, (e^{\varepsilon_1(\infty)+\varepsilon_2(\infty)} - 1)^2\}\right\}\right.$$
$$\left. + \sum_{j=3}^{\alpha} q^j \binom{\alpha}{j} e^{(j-1)(\varepsilon_1(j)+\varepsilon_2(j))} \min\{2, (e^{\varepsilon_1(\infty)+\varepsilon_2(\infty)} - 1)^j\}\right).$$

*Proof.* Given $(\alpha, \varepsilon_1 + \varepsilon_2)$-RDP in one round with sampling rate $q$, we can use Theorem 6 as

$$\varepsilon(\alpha) \leq \frac{1}{\alpha - 1} \cdot \log\left(1 + q^2 \binom{\alpha}{2} \min\left\{4(e^{\varepsilon_1(2)+\varepsilon_2(2)} - 1), \quad e^{\varepsilon_1(2)+\varepsilon_2(2)} \min\{2, (e^{\varepsilon_1(\infty)+\varepsilon_2(\infty)} - 1)^2\}\right\}\right.$$
$$\left. + \sum_{j=3}^{\alpha} q^j \binom{\alpha}{j} e^{(j-1)(\varepsilon_1(j)+\varepsilon_2(j))} \min\{2, (e^{\varepsilon_1(\infty)+\varepsilon_2(\infty)} - 1)^j\}\right). \tag{8}$$

Then, we compose $\varepsilon_q$ over $T$ rounds still using adaptive composition as in Theorem 4 as

$$\varepsilon(\alpha) \leq \frac{T}{\alpha - 1} \cdot \log\left(1 + q^2 \binom{\alpha}{2} \min\left\{4(e^{\varepsilon_1(2)+\varepsilon_2(2)} - 1), \quad e^{\varepsilon_1(2)+\varepsilon_2(2)} \min\{2, (e^{\varepsilon_1(\infty)+\varepsilon_2(\infty)} - 1)^2\}\right\}\right.$$
$$\left. + \sum_{j=3}^{\alpha} q^j \binom{\alpha}{j} e^{(j-1)(\varepsilon_1(j)+\varepsilon_2(j))} \min\{2, (e^{\varepsilon_1(\infty)+\varepsilon_2(\infty)} - 1)^j\}\right). \tag{9}$$

$\square$

Finally, we can convert the $(\alpha, \varepsilon(\alpha))$-RDP bound DP using Theorem 7. We have that for any $\delta \in (0, 1)$ (we used $0.001 < 1/M$ in our experiments), our algorithm satisfies $(\varepsilon, \delta)$-DP where

$$\varepsilon = \varepsilon(\alpha) + \frac{\log 1/\delta}{\alpha - 1}$$
$$\leq \frac{T}{\alpha - 1} \cdot \log\left(1 + q^2 \binom{\alpha}{2} \min\left\{4(e^{\varepsilon_1(2)+\varepsilon_2(2)} - 1), \quad e^{\varepsilon_1(2)+\varepsilon_2(2)} \min\{2, (e^{\varepsilon_1(\infty)+\varepsilon_2(\infty)} - 1)^2\}\right\}\right.$$
$$\left. + \sum_{j=3}^{\alpha} q^j \binom{\alpha}{j} e^{(j-1)(\varepsilon_1(j)+\varepsilon_2(j))} \min\{2, (e^{\varepsilon_1(\infty)+\varepsilon_2(\infty)} - 1)^j\}\right). \tag{10}$$

## B  Convergence Analysis Details

### B.1  Assumptions

Here we state the assumptions used in our convergence analysis.

**Assumption 1.** *For every $j \in [k]$, $F^j(\cdot)$ is $\lambda$-strongly convex and $L$-smooth.*

**Assumption 2.** *For every $\theta$ and every $j \in [k]$, the variance of $f(\theta; z)$ is upper bounded by $\eta^2$, when $z$ is sampled from $\mathcal{D}_j$, i.e., $\mathbb{E}_{z \sim \mathcal{D}_j}\left[(f(\theta, z) - F^j(\theta))^2\right] \leq \eta^2$.*

**Assumption 3.** *For every $\theta$ and every $j \in [k]$, the variance of $\nabla f(\theta; z)$ is upper bounded by $v^2$, where $z$ is sampled from $\mathcal{D}_j$, i.e., $\mathbb{E}_{z \sim \mathcal{D}_j}\left[\|\nabla f(\theta, z) - \nabla F^j(\theta)\|^2\right] \leq v^2$.*

**Assumption 4.** *For every cluster $S_j$ ($j \in [k]$) in any round, the variance of cluster sizes $|S_j|$ is upper bounded by $\mu^2$, i.e., $\mathbb{E}[(|S_j| - \frac{1}{k} \sum_{j \in [k]} |S_j|)^2] \leq \mu^2$.*

**Assumption 5.** *For every $\theta$ and every $j \in [k]$, the norm of the stochastic gradient $\nabla f(\theta; z)$ is bounded, i.e., there exists a constant $G > 0$ such that $\|\nabla f(\theta; z)\| \leq G, \forall z \sim \mathcal{D}_j$.*

Here, we further assume that the adopted DP clipping threshold $C_\theta$ is larger than any stocastic gradient norm, i.e., $C_\theta > G$, the clipping does not affect model update in our analysis. This assumption is commonly used across various differential privacy analysis (Li et al., 2022; Shi et al., 2023; Li et al., 2020c).

**Assumption 6.** *We assume that there exists $\alpha_0 \in (0, \frac{1}{2})$ s.t. the initial parameters $\theta_j^0$ satisfy $\|\theta_j^0 - \theta_j^*\| \leq (\frac{1}{2} - \alpha_0)\sqrt{\frac{\lambda}{L}}\Delta$ for every $j \in [k]$, where $\Delta := \min_{j \neq j'} \|\theta_j^* - \theta_{j'}^*\| \geq \mathcal{O}\left(\alpha_0^{\frac{1}{2}}k^{\frac{3}{4}}M^{-1}B^{-\frac{1}{2}}\right)$, $M > \left(\frac{B\Delta}{2\tau} - \frac{\rho}{\tau\gamma L}\right)$, and $\rho := M - (k-1)B$.*

Assumption 6 is a bounded-initialization (basin) condition, not a near-optimality requirement: the admissible initialization error is a constant fraction of the separation $\Delta$, the identifiability threshold of the problem. The same form is standard in analyses of IFCA (Ghosh et al., 2020), EM (Dempster et al., 1977; Balakrishnan et al., 2017), and Lloyd's $k$-means (Lloyd, 1982; Lu & Zhou, 2016); all our experiments use random initialization. The condition on $\Delta$ indicates well-separated underlying cluster models. We note that the assumption on $M$ can be satisfied when the client population is large.

### B.2  Proof of Lemma 1

*Proof.* Suppose we have cluster models $\{\theta_j\}_{j\in[k]}$, and assume that after certain steps, we have $\|\theta_j - \theta_j^*\| \leq (\frac{1}{2} - \beta)\sqrt{\frac{\lambda}{L}}\Delta, j \in [k]$. We first analyze the probability of incorrectly clustering a client taking into consideration the following three factors.

- Factor 1. Clients selecting their clusters by calculating training losses can introduce some inherent clustering error. We denote the probability of such incorrect clustering as $\Pr(E)$.

- Factor 2. Clients privatizing cluster identifiers introduces additional error, where the probability is denoted as $\Pr(H)$.

- Factor 3. `RR-Cluster` resamples updates from large to small groups, which also increases clustering error, denoted as $\Pr(O)$.

**Factor 1**: We begin with the definition of $E_i^{j,j'}$, which denotes an event where in a certain global round, client $i$ who belongs to cluster $j$ by ground truth, however been changed its cluster assignment to cluster $j'(j' \neq j)$, due to the min-loss cluster selection strategy. We bound this probability of erroneous cluster selection from Factor 1 as follows. Without loss of generality, we consider $E_i^{1,j}$. We have

$$\Pr\left(E_i^{1,j}\right) \leq \Pr\left(F_i(\theta_1) \geq F_i(\theta_j)\right) \leq \Pr\left(F_i(\theta_1) > t\right) + \Pr\left(F_i(\theta_j) \leq t\right) \tag{11}$$

for all $t \geq 0$. Choosing $t = \frac{F^1(\theta_1) + F^1(\theta_j)}{2}$, we have

$$\begin{aligned}
\Pr(F_i(\theta_1) > t) &= \Pr\left(F_i(\theta_1) > \frac{F^1(\theta_1) + F^1(\theta_j)}{2}\right) \\
&= \Pr\left(F_i(\theta_1) - F^1(\theta_1) > \frac{F^1(\theta_j) - F^1(\theta_1)}{2}\right)
\end{aligned} \tag{12}$$

for the first term. Considering that $F^1$ is $\lambda$-strongly convex and $L$-smooth and the assumption that $\|\theta_j^t - \theta_j^*\| \leq (\frac{1}{2} - \beta)\sqrt{\frac{\lambda}{L}}\Delta$ where $\Delta := \min_{j \neq j'} \|\theta_j^* - \theta_{j'}^*\|$, we have

$$F^1(\theta_j) \geq F^1(\theta_1^*) + \frac{\lambda}{2}\|\theta_j - \theta_1^*\|^2 \geq F^1(\theta_1^*) + \frac{\lambda\Delta^2}{2}(\frac{1}{2} + \beta)^2 \tag{13}$$

and

$$F^1(\theta_1) \leq F^1(\theta_1^*) + \frac{L}{2}\|\theta_1 - \theta_1^*\|^2 \leq F^1(\theta_1^*) + \frac{\lambda\Delta^2}{2}(\frac{1}{2} - \beta)^2. \tag{14}$$

Thus we have that

$$\frac{F^1(\theta_j) - F^1(\theta_1)}{2} \geq \frac{\lambda\Delta^2}{2}(\frac{1}{2} + \beta)^2 - \frac{\lambda\Delta^2}{2}(\frac{1}{2} - \beta)^2 = \frac{\beta\lambda\Delta^2}{2}. \tag{15}$$

According to the Chebyshev inequality and the bounded variance assumption, we obtain $\Pr(F_i(\theta_1) > t) \leq \frac{4\eta^2}{\beta^2\lambda^2\Delta^4}$ and similarly, $\Pr(F_i(\theta_j) \leq t) \leq \frac{4\eta^2}{\beta^2\lambda^2\Delta^4}$, and thus $\Pr(E_i^{j,j'}) \leq \frac{8\eta^2}{\beta^2\lambda^2\Delta^4}$.

**Factor 2**: We analyze the probability that adding random Gaussian noise to the one-hot cluster identifiers changes the index with the maximum value. Let $\sigma_s^2$ denote the variance of the Gaussian noise. We consider the event $H_i^{j,j'}$, which means client $i$ is assigned to some cluster $j$ ($j \in [k]$) by the min-loss cluster selection strategy, but is changed into cluster $j'$ due to the added noise to guarantee differential privacy. In this step, we add Gaussian noise with variance $\sigma_s^2$ to each coordinate of the one-hot $s_i$, resulting in $\tilde{s}_i$. Therefore,

$$\Pr(H_i^{j,j'}) \leq \Pr\left(1 + \mathcal{N}(0, \sigma_s^2) \leq 0 + \mathcal{N}(0, \sigma_s^2)\right) = \Pr(\mathcal{N}(0, 2\sigma_s^2) \leq -1). \tag{16}$$

Using the tail bound of Gaussian distributions $\Pr(x \geq t) \leq \frac{\sigma_s}{t\sqrt{2\pi}}\exp(-t^2/2\sigma_s^2)$, we have

$$\Pr(H_i^{j,j'}) \leq \frac{\sigma_s}{\sqrt{\pi}}\exp(-1/4\sigma_s^2). \tag{17}$$

**Factor 3**: We finally analyze the probability of event $O_i^{j,j'}$, where client $i$ is assigned to cluster $j$ after the previous two steps, but the assignment gets altered due to random rebalancing. Generally, if the mapping between a model update and a cluster has changed, then the model update should be in a large cluster $S_j, j \in [j_l]$. Assume that there are $h$ updates in a large cluster that are needed to merge into small clusters, we have

$$\Pr(O_i^{j,j'}) \leq \Pr(|S_j| > B) \cdot \frac{h}{\sum_{j \in j_l}|S_j|}. \tag{18}$$

For large clusters, considering the average cluster scale is $M/k$ by definition, and the variance of cluster scale is bounded by $\mu^2$. With one-sided Chebyshev inequality, we can have

$$\Pr(|S_j| > B) \leq \left(\frac{\mu}{M/k - B}\right)^2. \tag{19}$$

Combined with the facts that $h < M$ and $\sum_{j \in j_l}|S_j| > M/k$, it holds that

$$\Pr(O_i^{j,j'}) \leq \left(\frac{\mu}{M/k - B}\right)^2 \cdot \frac{M}{M/k} = \frac{k\mu^2}{(M/k - B)^2}. \tag{20}$$

If the final cluster assignment is incorrect, then the error must have occurred in at least one of the three steps. Hence, we can bound the error clustering rate $\tau$ as

$$\tau_i^{j,j'} \leq \Pr(E_i^{j,j'}) + \Pr(H_i^{j,j'}) + \Pr(O_i^{j,j'}) \leq \frac{8\eta^2}{\beta^2\lambda^2\Delta^4} + \frac{\sigma_s}{\sqrt{\pi}}\exp(-1/4\sigma_s^2) + \frac{k\mu^2}{(M/k - B)^2}. \tag{21}$$

Note that $\tau_i^{j,j'}$ denotes the probability of client update $i$ being clustered into any single cluster $j'$. We can get the probability of any client update $i$ been wrongly clustered into any other cluster as

$$\tau = \bigcup_{j' \in [k]} \{\tau_i^{j,j'}\} \leq k \cdot \tau_i^{j,j'} \leq \frac{8\eta^2 k}{\beta^2\lambda^2\Delta^4} + \frac{\sigma_s k}{\sqrt{\pi}}\exp(-1/4\sigma_s^2) + \frac{k^2\mu^2}{(M/k - B)^2}. \tag{22}$$

$\square$

### B.3 Proof of Theorem 2

*Proof.* Throughout this proof, $B$ denotes the enforced minimum occupancy with $1 \leq B \leq M/k$ (full participation assumed); see Remark 1 for the recovery of the base method via $B_0$. Suppose that at a certain communication round, we have that $\|\theta_j - \theta_j^*\| \leq (\frac{1}{2} - \beta)\sqrt{\frac{\lambda}{L}}\Delta$, for all $j \in [k]$. With out loss of generality, we focus on the update of cluster 1. Based on the updating rule, we have

$$\left\|\theta_1^+ - \theta_1^*\right\| = \left\|\theta_1 - \theta_1^* - \frac{\gamma}{|S_1|}\left(\sum_{i \in S_1}\nabla F_i(\theta_1) + \mathcal{N}(0, (\sqrt{5}C_\theta)^2\sigma_\theta^2 I_d)\right)\right\| \tag{23}$$

where $F_i(\theta) = F(\theta, D_i)$ with $D_i$ being the set of data from the $i$-th client used to compute the gradient. For cluster $j$, we denote its ground-truth client set as $S_j^*$, and the complement set as $\overline{S_j^*}$. Since $S_1 = (S_1 \cap S_1^*) \cup (S_1 \cap \overline{S_1^*})$ , we have

$$\|\theta_1^+ - \theta_1^*\| = \|\underbrace{\theta_1 - \theta_1^* - \frac{\gamma}{|S_1|}\sum_{i \in S_1 \cap S^*}\nabla F_i(\theta_1)}_{T_1} - \underbrace{\frac{\gamma}{|S_1|}\sum_{i \in S_1 \cap \overline{S_i^*}}\nabla F_i(\theta_1)}_{T_2} - \underbrace{\frac{\gamma}{|S_1|}\mathcal{N}(0, (\sqrt{5}\sigma_\theta C_\theta)^2 I_d)}_{T_3}\|. \quad (24)$$

Using triangle inequality, we obtain $\|\theta_1^+ - \theta_1^*\| \le \|T_1\| + \|T_2\| + \|T_3\|$.

**Bound** $\|T_1\|$: We split $T_1$ as follows:

$$T_1 = \underbrace{\theta_1 - \theta_1^* - \widehat{\gamma}\nabla F^1(\theta_1)}_{T_{11}} + \underbrace{\widehat{\gamma}(\nabla F^1(\theta_1) - \frac{1}{|S_1 \cap S_1^*|}\sum_{i \in S_1 \cap S_1^*}\nabla F_i(\theta_1))}_{T_{12}}, \quad (25)$$

where $\widehat{\gamma} := \frac{\gamma|S_1 \cap S_1^*|}{|S_1|}$. Based on $\lambda$-strongly convexity and $L$-smoothness of $F^1$, we can bound $\|T_{11}\|$ as

$$\|T_{11}\| = \|\theta_1 - \theta_1^* - \widehat{\gamma}\nabla F^1(\theta_1)\| \le \left(1 - \frac{\widehat{\gamma}\lambda L}{\lambda + L}\right)\|\theta_1 - \theta_1^*\| \quad (26)$$

when $\widehat{\gamma} < \frac{1}{L}$. For $T_{12}$, we have $\mathbb{E}[\|T_{12}\|^2] = \frac{v^2}{|S_1 \cap S_1^*|}$, which means $E[\|T_{12}\|] \le \frac{v}{\sqrt{|S^1 \cap S_1^*|}}$. We can then bound the $\|T_{12}\|$ by Markov inequality as: for any $\delta_1 \in (0, 1]$, we have with probability at least $1 - \delta_1$,

$$\|T_{12}\| \le \frac{v}{\delta_1\sqrt{|S_1 \cap S_1^*|}}. \quad (27)$$

Taking into accountant $T_{11}$, $T_{12}$ and fact that $\lambda < L$, we have with probability at least $1 - \delta_1$,

$$\|T_1\| \le \left(1 - \frac{\gamma L|S_1 \cap S_1^*|}{2|S_1|}\right)\|\theta_1 - \theta_1^*\| + \frac{v}{\delta_1\sqrt{|S_1 \cap S_1^*|}}. \quad (28)$$

**Bound** $\|T_2\|$: We propose to split the $(S_1 \cap \overline{S_1^*})$ into $\bigcup_{j \ne 1, j \in [k]}(S_1 \cap S_j^*)$. Without loss of generality, we first analyze the $(S_1 \cap S_j^*)$ as

$$T_{2j} = |S_1 \cap S_j^*|\underbrace{\nabla F^j(\theta_1)}_{T_{21j}} + \underbrace{\sum_{i \in S_i \cap S_j^*}(\nabla F_i(\theta_1) - \nabla F^j(\theta_1))}_{T_{22j}}, \quad (29)$$

where $T_2 = \frac{\gamma}{|S_1|}\sum_{j \in [k]}T_{2j}$ About $T_{21j}$, we have

$$\|T_{21j}\| = \|\nabla F^j(\theta_1) - \nabla F^j(\theta_j^*)\| \le L\|\theta_1 - \theta_j^*\|$$
$$\le L\|\theta_1 - \theta_1^*\| + L\|\theta_1^* - \theta_j^*\| \le \frac{3}{2}L\Delta. \quad (30)$$

Further, we have $\mathbb{E}\left[\|T_{22j}\|^2\right] = |S_1 \cap S_j^*|v^2$, which implies $\mathbb{E}[\|T_{22j}\|] \le \sqrt{|S_1 \cap S_j^*|}v$. And according to Markov inequality, for any $\delta_2 \in [0, 1]$, with probability at least $1 - \delta_2$, we have

$$\|T_{22j}\| = \left\|\sum_{i \in S_1 \cap S_j^*}\nabla F_i(\theta_1) - \nabla F^j(\theta_1)\right\| \le \frac{\sqrt{|S_1 \cap S_j^*|}v}{\delta_2}. \quad (31)$$

Considering there are $k$ clusters, we have with probability at least $(1 - k\delta_2)$,

$$\|T_2\| \le \frac{3\gamma L\Delta}{2|S_1|}|S_1 \cap \overline{S_1^*}| + \frac{\gamma v\sqrt{k}}{\delta_2|S_1|}\sqrt{|S_1 \cap \overline{S_1^*}|}. \quad (32)$$

**Bound** $\|T_3\|$: We give the norm bound of the gaussian noise from DP as follows. $\|T_3\| = \|\frac{\gamma}{|S_1|}\mathcal{N}(0, (\sqrt{5}\sigma_\theta C_\theta)^2 I_d)\|$, which is a vector with dimension of model parameters $d$ and scale of DP noise

multiplier. First, we have

$$\|T_3\| = \big\|\frac{\gamma}{|S_1|}\mathcal{N}(0,(\sqrt{5}\sigma_\theta C_\theta)^2 I_d)\big\| \leq \frac{\sqrt{5}\gamma\sigma_\theta C_\theta}{B}\|\mathcal{N}(0,I_d)\|. \tag{33}$$

Denote $X \sim \mathcal{N}(0,I_d)$. Then $\|X\|^2$ follows a $\chi^2$-distribution with $d$ degrees of freedom, i.e. $\|X\|^2 \sim \chi_d^2$. We then use the upper bound for a $\chi^2$-tail to bound $\|T_3\|$ as

$$P\left(\|X\|^2 \leq d + 2\sqrt{d\ln(\frac{1}{\delta_3})} + 2\ln(\frac{1}{\delta_3})\right) \geq 1 - \delta_3. \tag{34}$$

Multiplying both sides by $(\sigma_\theta/B)$ and taking the square root yields

$$P\left(\|T_3\| \leq \frac{\sqrt{5}\gamma\sigma_\theta C_\theta}{B}\sqrt{d + 2\sqrt{d\ln(\frac{1}{\delta_3})} + 2\ln(\frac{1}{\delta_3})}\right) \geq 1 - \delta_3. \tag{35}$$

Which means that we have probability at least $1 - \delta_3$ that

$$\|T_3\| \leq \frac{\sqrt{5}\gamma\sigma_\theta C_\theta}{B}\sqrt{d + 2\sqrt{d\ln(\frac{1}{\delta_3})} + 2\ln(\frac{1}{\delta_3})}. \tag{36}$$

With bounded $\|T_1\|$ and $\|T_2\|$ above, we have probability at least $1 - (\delta_1 + k\delta_2 + \delta_3)$ that

$$\|\theta_1^+ - \theta_1^*\| \leq \left(1 - \frac{\gamma L|S_1 \cap S_1^*|}{2|S_1|}\right)\|\theta_1 - \theta_1^*\|$$
$$+ \frac{v}{\delta_1\sqrt{|S_1 \cap S_1^*|}} + \frac{3\gamma L\Delta}{2|S_1|}|S_1 \cap \overline{S_1^*}|$$
$$+ \frac{\gamma v\sqrt{k}}{\delta_2|S_1|}\sqrt{|S_1 \cap \overline{S_1^*}|} + \frac{\sqrt{5}\gamma\sigma_\theta C_\theta}{B}\sqrt{d + 2\sqrt{d\ln(\frac{1}{\delta_3})} + 2\ln(\frac{1}{\delta_3})}. \tag{37}$$

We bound $|S_1|$, $|S_1 \cap S_1^*|$ and $|S_1 \cap \overline{S_1^*}|$ next to complete the theorem. Based on the updating rules of our method, we note that the maximum cardinality of any $S_j$ is bounded by $\rho := M - (k-1)B$, thus we have $|S_1| \leq \rho$. Also, after rebalancing, we have $|S_1| \geq B$. Given the error rate in clustering from Lemma 1, we have $\mathbb{E}[|S_1 \cap \overline{S_1^*}|] \leq \tau M$, plus the Markov inequality, we obtain probability at least $1 - \delta_4$ that $|S_1 \cap \overline{S_1^*}| \leq \tau M/\delta_4$, meanwhile, we have $|S_1 \cap S_1^*| = |S_1| - (|S_1 \cap \overline{S_1^*}|) \geq (B - \frac{\tau M}{\delta_4})$. Let $\delta_c \geq \delta_1 + k\delta_2 + \delta_3 + \delta_4$ and choose $\delta_1 = \delta_c/4, \delta_2 = \delta_c/4k, \delta_3 = \delta_c/4, \delta_4 = \delta_c/4$, then the failure probability can be bounded by $\delta_c$. Finally, we have with probability at least $(1 - \delta_c)$, it holds that

$$\|\theta_1^+ - \theta_1^*\| \leq \left(1 - \frac{\gamma L(B - \frac{2\tau M}{\delta_c})}{2\rho}\right)\|\theta_1 - \theta_1^*\|$$
$$+ \frac{4v}{\sqrt{\delta_c(B\delta_c - 4\tau M)}} + \frac{6\tau\gamma L\Delta M}{\delta_c B} + \frac{8\gamma vk\sqrt{\tau kM}}{B\delta_c\sqrt{\delta_c}} + \frac{\sqrt{5}\gamma\sigma_\theta C_\theta}{B}\sqrt{d + 2\sqrt{d\ln(\frac{4}{\delta_c})} + 2\ln(\frac{4}{\delta_c})}, \tag{38}$$

where

$$\rho = (M - (k-1)B), \text{ and } \tau = \frac{8\eta^2 k}{\beta^2\lambda^2\Delta^4} + \frac{\sigma_s k}{\sqrt{\pi}}\exp(-1/4\sigma_s^2) + \frac{k^2\mu^2}{(M/k - B)^2}. \tag{39}$$

$\square$

## B.4 Overall Convergence of `RR-Cluster` over $T$ Rounds

**Corollary 1.** *Suppose Assumption 1-6 hold and $1 \leq B \leq M/k$ (the enforced-occupancy regime), after*

$T = \frac{1}{K}\log\left(\frac{\Delta\sqrt{\lambda}}{2\epsilon_T\sqrt{L}}\right) + \frac{\log\left(\frac{\Delta\sqrt{\lambda}}{8(\frac{1}{2} - \alpha_0)\sqrt{L}}\right)}{\log(1-K)}$ *rounds, for all cluster $j \in [k]$, we have at least probability $(1 - \delta_c')$, $\|\theta_j^T - \theta_j^*\| \leq \epsilon_T$, where*

$$\epsilon_T = \frac{2\epsilon}{K}, \ K = \frac{\gamma L(B - 2\tau M/\delta_c)}{2\rho}, \ \delta_c' = kT\delta_c.$$

*Proof.* Recall that the error floor of Theorem 2 $\epsilon$ is

$$\epsilon = \frac{4v}{\sqrt{\delta_c(B\delta_c - 4\tau M)}} + \frac{6\tau\gamma L\Delta M}{\delta_c B} + \frac{8\gamma vk\sqrt{\tau kM}}{B\delta_c\sqrt{\delta_c}} + \frac{\sqrt{5}\gamma\sigma_\theta C_\theta}{B}\sqrt{d + 2\sqrt{d\ln(\frac{4}{\delta_c})} + 2\ln(\frac{4}{\delta_c})}. \tag{40}$$

Let $K$ be the subtraction factor $\frac{\gamma L(B - 2\tau M/\delta_c)}{2\rho}$ in Eq. equation 38. We rewrite Eq. equation 38 as

$$\|\theta_1^+ - \theta_1^*\| \le (1 - K)\|\theta_1 - \theta_1^*\| + \epsilon = \|\theta_1 - \theta_1^*\| + (\epsilon - K\|\theta_1 - \theta_1^*\|). \tag{41}$$

Given the bound of $\Delta$ in Assumption 6, we can ensure $\epsilon \le K(\frac{1}{2} - \alpha_0)\Delta$, thus we have that the proposed method is contractive, that is, $\|\theta_1^{t+1} - \theta_1^*\| \le \|\theta_1^t - \theta_1^*\|$.

In the $t$-th round, assume we have $\|\theta_1^t - \theta_1^*\| \le (\frac{1}{2} - \alpha_t)\sqrt{\frac{\lambda}{L}}\Delta$, where the sequence of $\alpha_t$ should be non-decreasing due to its contractive convergence.

Suppose after $T'$ rounds we have $\alpha_t \le \frac{1}{4}$, this can be achieved if both

$$(1 - K)^{T'}(\frac{1}{2} - \alpha_0)\Delta \le \frac{1}{8}\sqrt{\frac{\lambda}{L}}\Delta \text{ , and } \frac{1}{K}\epsilon \le \frac{1}{8}\sqrt{\frac{\lambda}{L}}\Delta \tag{42}$$

hold, where the latter equation holds given bounds on $\Delta$ in Assumption 6. Solving Eq. equation 42, we can get

$$T' \ge \frac{\log\left(\frac{\Delta\sqrt{\lambda}}{8(\frac{1}{2} - \alpha_0)\sqrt{L}}\right)}{\log(1 - K)}, \tag{43}$$

which means that after $T'$ rounds, $\|\theta_1^{T'} - \theta_1^*\| \le \frac{1}{4}\sqrt{\frac{\lambda}{L}}\Delta$.

Let $T = T' + T''$, after $T$ rounds in total (with another $T''$ rounds), we have probability at least $1 - kT\delta_c$, for all $j \in [k]$

$$\|\theta_1^T - \theta_1^*\| \le (1 - K)^{T''}\|\theta_1^{T'} - \theta_1^*\| + \frac{1}{K}\epsilon, \tag{44}$$

where the last term $\frac{\epsilon}{K}$ is calculated by the sum of series of $\epsilon + (1 - K)\epsilon + (1 - K)^2\epsilon + \cdots$.

Let $T'' \ge \frac{1}{K}\log\left(\frac{k\Delta\sqrt{\lambda}}{4\epsilon\sqrt{L}}\right)$, we can bound $(1 - K)^{T''}\|\theta_1^{T'} - \theta_1^*\|$ as

$$(1 - K)^{T''}\|\theta_1^{T'} - \theta_1^*\| \le e^{-KT''} \cdot \frac{1}{4}\sqrt{\frac{\lambda}{L}}\Delta \le \frac{1}{K}\epsilon, \tag{45}$$

which means $\|\theta_1^T - \theta_1^*\| \le \frac{2\epsilon}{K} = \epsilon_T$ after $T = \frac{1}{K}\log\left(\frac{K\Delta\sqrt{\lambda}}{4\epsilon\sqrt{L}}\right) + \frac{\log\left(\frac{\Delta\sqrt{\lambda}}{8(\frac{1}{2} - \alpha_0)\sqrt{L}}\right)}{\log(1 - K)} = \frac{1}{K}\log\left(\frac{\Delta\sqrt{\lambda}}{2\epsilon_T\sqrt{L}}\right) + \frac{\log\left(\frac{\Delta\sqrt{\lambda}}{8(\frac{1}{2} - \alpha_0)\sqrt{L}}\right)}{\log(1 - K)}$ rounds, with probability at least $(1 - \delta_c')$ where $\delta_c' = kT\delta_c$.

$\square$

## C  Experimental Details

**Models.**  For the image classification tasks, we employed a convolutional neural network, and for the Shakespeare classification task, we used a LSTM network for classification. For synthetic data, we used a linear regression model with slope and intersect as trainable parameters. For the larger-backbone study in Appendix D.5, we use ResNet-18 and ViT-B initialized from public ImageNet pre-training.

**Hyperparameters.**  For the clipping bound of model updates $C_\theta$, we conduct grid search in `np.logspace(-1,-3,5)` and select the value based on model accuracy on validation set for all methods. For our hyperparameter $B$ (minimal cluster size), if not specified, we use grid search $B$ in $\{4, 8, 12\}$ by default in all experiments. For the clipping threshold of cluster identifiers $s$, considering that all $s$ are one-hot vectors, the clipping threshold cannot affect cluster assignment. Since $\tilde{s} = s/\max(1, \frac{s}{C_s}) + \mathcal{N}(0, C_s^2) = C_s \cdot (s + \mathcal{N}(0, 1))$

if we select $C_s \leq 1$, and that server assign clusters based on the position of $\max\{\widetilde{s}_j\}_{j\in k}$, so it is only the position of $\max\{\widetilde{s}_j\}_{j\in k}$ that matters, which is irrelevant to $C_s$. In experiments, we use 0.1 as the $C_s$. We tune local client-side learning rate from {1e-6, 1e-5, 1e-4, 1e-3}, and the number of local epochs from {5, 10, 15, 20} on DP-FedAvg (McMahan et al., 2018) and use the same values for all methods.

**Hardware.** Most experiments are conducted using 2 NVIDIA L40 GPU on a AMD Thread-Ripper HEDT platform in a desktop.

# D    Additional Experimental Results

## D.1    Fine-grained Accuracy Metrics with Max/Min Accuracy

To provide a more comprehensive breakdown of model behavior under differential privacy, we present a unified table containing *average*, *maximum*, and *minimum* client accuracy in Table 7.

| Methods | $\varepsilon = 2$ | | | $\varepsilon = 4$ | | | $\varepsilon = 8$ | | |
|---|---|---|---|---|---|---|---|---|---|
| | Avg | Max | Min | Avg | Max | Min | Avg | Max | Min |
| DP-FedAvg | 60.88 | 69.20 | 41.94 | 61.50 | 69.38 | 45.71 | 62.72 | 71.14 | 48.65 |
| DP-IFCA | 62.68 | 75.85 | 08.82 | 64.46 | 76.73 | 21.00 | 65.35 | 79.55 | 22.12 |
| DP-FeSEM | 63.16 | 74.40 | 16.12 | 64.68 | 77.76 | 18.42 | 64.76 | 78.44 | 24.32 |
| DP-FedCAM | 49.70 | 57.74 | 18.33 | 56.86 | 66.48 | 21.43 | 61.54 | 72.32 | 29.44 |
| RR-Cluster (IFCA) | 64.92 | 80.10 | 21.72 | 66.26 | 80.75 | 34.61 | 67.48 | 82.92 | 42.36 |

Table 7: Fine-grained comparison of client-wise performance, including average, maximum and minimum client accuracy under different privacy budgets.

**Discussion.** By consolidating average, maximum, and minimum accuracy into a single table, we highlight how RR-Cluster improves client-wise performance across a wide spectrum of client conditions.

## D.2    Full results with Federated Personalization Methods

| Methods | Balanced Clusters | | | Imbalanced Clusters | | |
|---|---|---|---|---|---|---|
| | $\varepsilon = 2$ | $\varepsilon = 4$ | $\varepsilon = 8$ | $\varepsilon = 2$ | $\varepsilon = 4$ | $\varepsilon = 8$ |
| DP-FedAvg | 60.88 | 61.50 | 62.72 | 60.80 | 61.53 | 61.59 |
| DP-FedPer | 61.03 | 61.78 | 62.19 | 60.42 | 61.83 | 62.35 |
| DP-IFCA | 62.68 | 64.46 | 65.35 | 56.12 | 58.05 | 59.63 |
| DP-FeSEM | 63.16 | 64.68 | 64.76 | 58.55 | 59.82 | 60.10 |
| DP-FedCAM | 49.70 | 56.86 | 61.54 | 43.98 | 47.51 | 50.35 |
| DP-CFL | 61.62 | 62.73 | 65.66 | 58.51 | 60.54 | 61.67 |
| DP-GradPart | 63.39 | 64.88 | 65.17 | 60.68 | 61.23 | 62.11 |
| DP-FLUX | 61.06 | 61.40 | 63.52 | 58.78 | 59.41 | 60.95 |
| RR-Cluster (IFCA) | 64.92 $_{\uparrow 2.24}$ | 66.26 $_{\uparrow 1.80}$ | 67.48 $_{\uparrow 2.13}$ | 61.06 $_{\uparrow 4.94}$ | 61.64 $_{\uparrow 3.59}$ | 62.75 $_{\uparrow 3.12}$ |
| RR-Cluster (FeSEM) | 63.38 $_{\uparrow 0.22}$ | 64.78 $_{\uparrow 0.10}$ | 64.80 $_{\uparrow 0.04}$ | 58.64 $_{\uparrow 0.09}$ | 60.12 $_{\uparrow 0.30}$ | 60.65 $_{\uparrow 0.55}$ |
| RR-Cluster (FedCAM) | 51.23 $_{\uparrow 1.53}$ | 57.49 $_{\uparrow 0.63}$ | 63.73 $_{\uparrow 2.19}$ | 45.01 $_{\uparrow 1.03}$ | 47.89 $_{\uparrow 0.38}$ | 50.59 $_{\uparrow 0.24}$ |

Table 8: Full version of Table 1. Comparison with baselines on FashionMNIST. See discussions in Section 5.2.

In this section, we present the full versions of the experimental results tables discussed in Section 5.2 of the main paper. For detailed analysis and discussion of these findings, please refer to Section 5.2.

| Methods | Low Client Heterogeneity | | | High Client Heterogeneity | | |
|---|---|---|---|---|---|---|
| | $\varepsilon = 2$ | $\varepsilon = 4$ | $\varepsilon = 8$ | $\varepsilon = 2$ | $\varepsilon = 4$ | $\varepsilon = 8$ |
| DP-FedAvg | 26.47 | 30.95 | 32.61 | 23.82 | 24.46 | 25.54 |
| DP-FedPer | 40.67 | 40.81 | 41.23 | 24.78 | 25.51 | 26.12 |
| DP-IFCA | 62.23 | 65.39 | 65.93 | 32.53 | 35.60 | 36.19 |
| DP-FeSEM | 60.43 | 65.72 | 66.24 | 31.12 | 37.52 | 38.74 |
| DP-FedCAM | 57.35 | 57.22 | 61.26 | 23.66 | 24.55 | 29.70 |
| RR-Cluster (IFCA) | 64.96 ↑2.73 | 66.49 ↑0.77 | 66.87 ↑0.63 | 36.23 ↑3.70 | 37.69 ↑0.17 | 38.70 |

Table 9: Full version of Table 4. Comparison with baselines on EMNIST. See discussions in Section 5.2.

| Methods | $\varepsilon = 4$ | $\varepsilon = 8$ | $\varepsilon = 16$ |
|---|---|---|---|
| DP-FedAvg | 04.47 | 13.43 | 17.53 |
| DP-FedPer | 13.09 | 13.14 | 13.18 |
| DP-IFCA | 12.62 | 12.64 | 13.20 |
| DP-FeSEM | 10.97 | 12.99 | 15.89 |
| DP-FedCAM | - | - | 04.47 |
| RR-Cluster(IFCA) | 13.09 | 13.60 | 16.23 |

Table 10: Full version of Table 3. Comparison with baselines on Shakespeare. See discussions in Section 5.2.

| Methods | CIFAR10 | | CIFAR100 | | TinyImagenet | |
|---|---|---|---|---|---|---|
| | $\varepsilon = 4$ | $\varepsilon = 8$ | $\varepsilon = 4$ | $\varepsilon = 8$ | $\varepsilon = 4$ | $\varepsilon = 8$ |
| DP-FedAvg | 52.89 | 54.48 | 14.05 | 14.49 | 13.67 | 14.06 |
| DP-FedProx | 52.04 | 54.61 | 14.48 | 14.53 | 14.01 | 14.12 |
| DP-IFCA | 53.91 | 56.77 | 16.29 | 17.47 | 15.79 | 16.76 |
| DP-FeSEM | 53.01 | 55.87 | 16.23 | 17.08 | 15.65 | 16.44 |
| DP-FedCAM | 52.50 | 54.73 | 16.36 | 17.04 | 15.71 | 16.38 |
| RR-Cluster (IFCA) | 56.45 ↑2.54 | 58.22 ↑1.45 | 17.20 ↑0.84 | 17.93 ↑0.46 | 16.73 ↑0.94 | 17.52 ↑0.76 |

Table 11: Full version of Table 2. Comparison with baselines on CIFAR10, CIFAR100, and TinyImagenet. See discussions in Section 5.2.

### D.3 Additional Diagnostic Experiments

This section presents the three diagnostic experiments summarized in Section 5.3.

**Donor-Selection Strategies.** As discussed in Section 3.4, uniform donor selection consumes no extra privacy budget. We compare it with loss- and gradient-similarity selection, privatized under the same total budget (25%/75% split between selection signals and model updates) on FashionMNIST with balanced rotations:

| Methods | $\varepsilon = 2$ | $\varepsilon = 4$ | $\varepsilon = 8$ |
|---|---|---|---|
| RR-Cluster | **64.92** | 66.26 | 67.48 |
| RR-Cluster w/ Loss Sim. | 64.48 | 65.92 | 67.81 |
| RR-Cluster w/ Gradient Sim. | 64.72 | **66.59** | **67.90** |

The gains of the data-dependent variants are marginal, as their benefit is offset by the budget spent on the selection signals, and all variants outperform DP-IFCA (62.68 / 64.46 / 65.35). RR-Cluster is thus robust to the donor-selection strategy, and we adopt the uniform variant for its simplicity and zero privacy cost.

**Isolating the Rebalancing Bias.** To isolate the bias introduced purely by rebalancing, we run RR-Cluster on FashionMNIST with ground-truth (oracle) assignments and no privacy noise, so rebalancing is the only error source; the gap to IFCA under the same oracle assignments measures the pure rebalancing bias:

| IFCA (reference) | RR-Cluster $B = 2$ | $B = 4$ | $B = 6$ | $B = 8$ |
|---|---|---|---|---|
| 75.82 | 76.26 | 75.40 | 75.01 | 73.85 |

The bias stays small for small $B$ and grows as $B$ approaches the average cluster size, matching the $k^2\mu^2/(M/k-B)^2$ term of Lemma 1; $B = 2$ slightly outperforms the reference, echoing the collapse-mitigation side effect in Section 3.4.

**Per-Cluster Diagnostics under Imbalanced Clusters.** We report per-cluster accuracies under the imbalanced FashionMNIST setting at $\varepsilon = 2$ (majority = largest cluster, minority = smallest):

| Methods | Majority cluster | Minority cluster | Avg |
|---|---|---|---|
| DP-IFCA | 64.57 | 30.17 | 56.12 |
| RR-Cluster | **65.23** | **46.09** | **61.06** |

By enforcing at least $B$ updates per round for the minority cluster, RR-Cluster raises its accuracy by nearly 16 points without sacrificing the majority cluster.

## D.4 Mean and Standard Deviation of Main Results

We repeat the FashionMNIST experiments with 3 independent runs and report mean $\pm$ standard deviation in Table 12, corresponding to Table 1 in the main text. All standard deviations are below 0.5%, while the improvements of RR-Cluster over DP-IFCA range from 1.8 to 4.9 points, well beyond run-to-run variance.

| Methods | Balanced Clusters | | | Imbalanced Clusters | | |
|---|---|---|---|---|---|---|
| | $\varepsilon = 2$ | $\varepsilon = 4$ | $\varepsilon = 8$ | $\varepsilon = 2$ | $\varepsilon = 4$ | $\varepsilon = 8$ |
| DP-FedAvg | $60.88 \pm 0.23$ | $61.50 \pm 0.16$ | $62.72 \pm 0.13$ | $60.80 \pm 0.35$ | $61.53 \pm 0.18$ | $61.59 \pm 0.39$ |
| DP-IFCA | $62.68 \pm 0.21$ | $64.46 \pm 0.13$ | $65.35 \pm 0.08$ | $56.12 \pm 0.28$ | $58.05 \pm 0.27$ | $59.63 \pm 0.36$ |
| DP-FeSEM | $63.16 \pm 0.15$ | $64.68 \pm 0.05$ | $64.76 \pm 0.22$ | $58.55 \pm 0.17$ | $59.82 \pm 0.24$ | $60.10 \pm 0.19$ |
| DP-FedCAM | $49.70 \pm 0.28$ | $56.86 \pm 0.35$ | $61.54 \pm 0.30$ | $43.98 \pm 0.43$ | $47.51 \pm 0.13$ | $50.35 \pm 0.24$ |
| RR-Cluster (IFCA) | $64.92 \pm 0.14$ | $66.26 \pm 0.17$ | $67.48 \pm 0.11$ | $61.06 \pm 0.25$ | $61.64 \pm 0.23$ | $62.75 \pm 0.08$ |
| RR-Cluster (FeSEM) | $63.38 \pm 0.27$ | $64.78 \pm 0.19$ | $64.80 \pm 0.06$ | $58.64 \pm 0.32$ | $60.12 \pm 0.31$ | $60.65 \pm 0.13$ |
| RR-Cluster (FedCAM) | $51.23 \pm 0.22$ | $57.49 \pm 0.14$ | $63.73 \pm 0.30$ | $45.01 \pm 0.19$ | $47.89 \pm 0.15$ | $50.59 \pm 0.14$ |

Table 12: Mean $\pm$ standard deviation (3 runs) on FashionMNIST, corresponding to Table 1.

## D.5 Results with Larger Backbones (ResNet-18 and ViT-B)

We evaluate ResNet-18 and ViT-B on CIFAR10/100 under the "Balanced Clusters" setup of Table 2, both initialized from public ImageNet pre-training (no privacy cost, following standard DP practice) and privately fine-tuned under the same protocol and accounting. As shown in Table 13, RR-Cluster outperforms DP-FedAvg and DP-IFCA on both architectures and budgets, confirming that the benefit of the occupancy floor $B$ is architecture-agnostic.

| Methods | CIFAR10 | | | | CIFAR100 | | | |
|---|---|---|---|---|---|---|---|---|
| | ResNet-18 | | ViT-B | | ResNet-18 | | ViT-B | |
| | $\varepsilon = 4$ | $\varepsilon = 8$ | $\varepsilon = 4$ | $\varepsilon = 8$ | $\varepsilon = 4$ | $\varepsilon = 8$ | $\varepsilon = 4$ | $\varepsilon = 8$ |
| DP-FedAvg | 75.12 | 78.67 | 82.44 | 84.25 | 45.71 | 46.56 | 49.25 | 50.08 |
| DP-IFCA | 77.31 | 80.14 | 82.93 | 85.50 | 46.02 | 50.89 | 52.42 | 54.90 |
| RR-Cluster | **81.66** | **83.18** | **84.97** | **87.41** | **48.62** | **51.74** | **53.38** | **55.77** |

Table 13: Test accuracy on CIFAR10 and CIFAR100 with larger backbones (publicly pre-trained, privately fine-tuned).

# E    Table of Notations

Table 14 summarizes the notations used throughout the paper.

| Symbol | Description |
| --- | --- |
| $M$ | total number of clients |
| $k$ | number of clusters |
| $q$ | client sampling rate per round |
| $M^t$ | set of clients sampled at round $t$ |
| $T$ | total number of communication rounds |
| $B$ | rebalancing threshold (minimum per-cluster occupancy), $1 \leq B \leq qM/k$ |
| $B_0$ | realized minimum occupancy of the base method (Remark 1) |
| $S_j^t$ | set of client indices assigned to cluster $j$ at round $t$ |
| $j_l$, $j_s$ | index sets of large ($|S_j^t| \geq B$) and small clusters |
| $R_B$, $P$ | rebalancing map (Definition 3) and its donor pool |
| $D_i$, $F_i$ | local dataset and local empirical loss of client $i$ |
| $F^j$, $\theta_j^*$ | population loss and optimal model of cluster $j$ |
| $\theta_j^t$, $\Delta\theta_i^t$ | cluster model at round $t$, and model update of client $i$ |
| $s_i$, $\widetilde{s}_i$ | cluster identifier (one-hot) of client $i$, and its privatized version |
| $C_\theta$, $C_s$ | clipping bounds for model updates and cluster identifiers |
| $\sigma_\theta$, $\sigma_s$ | noise multipliers for model updates and cluster identifiers |
| $\varepsilon$, $\delta$ | differential privacy parameters |
| $\alpha$ | Rényi differential privacy order |
| $d$ | dimension of model parameters |
| $\gamma$ | server learning rate |
| $L$, $\lambda$ | smoothness and strong-convexity constants |
| $\eta^2$, $v^2$, $\mu^2$ | variance bounds on losses, gradients, and cluster sizes |
| $G$ | upper bound on stochastic gradient norms |
| $\Delta$ | minimum inter-cluster separation $\min_{j \neq j'} \|\theta_j^* - \theta_{j'}^*\|$ |
| $\tau$ | per-round misclustering probability (Lemma 1) |
| $\rho$ | maximum cluster cardinality, $\rho = M - (k-1)B$ |
| $\delta_c$ | failure probability in Theorem 2 |

Table 14: Table of notations.

