# OpenReview forum: "Differentially Private Federated Clustering with Random Rebalancing"
_TMLR — Under review for TMLR_

### Review · Reviewer_Qg4Q · 2026-06-04

**Summary Of Contributions:**

This paper proposes RR-Cluster, a federated clustering plugin that mitigates differential privacy utility degradation. Randomly rebalancing cluster assignments to ensure minimum sizes theoretically and empirically improves privacy-utility trade-offs. Specifically, this paper demonstrates a certain degree of innovation, the theoretical proofs are complete, and the experiments adequately support the results. However, I still have some concerns.  For example, this paper does not adequately discuss the latest research, and the selection of the baseline needs improvement.

**Audience:**

Yes

**Audience Explanation:**

Readers in fields such as federated learning, clustering research, and model safety will find the findings of this paper very interesting.

**Broader Impact Concerns:**

There are no significant ethical concerns associated with this work.

**Claims And Evidence:**

Yes

**Claims Explanation:**

The paper provides rigorous mathematical proofs, and although the experimental evaluations have minor flaws, they are comprehensive and sufficiently support the core conclusions.

**Requested Changes:**

1. The related work section lacks a discussion of the most recent research published between 2024 and 2026.
2. Consider adding a Table of Notations to summarize and explain the key variables and equations. This will greatly improve the readability of the mathematical formulations.
3. The experimental evaluation relies on baseline algorithms published up to 2023 (e.g., FedAvg, FedProx, FedPer, IFCA, FeSEM, and FedCAM). To fully establish the framework's current competitiveness, the comparative analysis must be updated to include recent state-of-the-art federated clustering methodologies published between 2024 and 2026.
4. The paper lacks guidance on setting the privacy budget $\epsilon$ in practical applications. Please discuss heuristic strategies for systematically selecting optimal $\epsilon$ values.
5. It would be highly beneficial to include a section on future work. Summarizing potential directions for future research is necessary to provide a complete perspective on the proposed method.

---

> ### Author Response · Authors · 2026-07-12
> **Rebuttal by Authors**
>
> Dear Reviewer Qg4Q:
>
> Thank you for the positive feedback. We address each requested change below.
>
> ---
>
> **[#1: Discussion of 2024–2026 research]**
>
> We have added a dedicated paragraph to `Section 2` in the revised paper discussing the most recent research: gradient-based partitioning CFL (Kim et al., ICML 2024), DPCFL (TMLR 2025), integrated CFL pipelines (Guo et al., ICLR 2025), FLUX (NeurIPS 2025), and time-adaptive privacy spending (ICLR 2025), each positioned by its privacy unit and its relation to our contribution.
>
> Beyond the discussion, we further include the three most recent clustered-FL methods as new baselines under our client-level DP protocol, on FashionMNIST with balanced and imbalanced image rotations:
>
> | FashionMNIST | Bal $\varepsilon=2$ | $\varepsilon=4$ | $\varepsilon=8$ | Imb $\varepsilon=2$ | $\varepsilon=4$ | $\varepsilon=8$ |
> |---|---|---|---|---|---|---|
> | DP-FedAvg | 60.88 | 61.50 | 62.72 | 60.80 | 61.53 | 61.59 |
> | DP-IFCA | 62.68 | 64.46 | 65.35 | 56.12 | 58.05 | 59.63 |
> | DP-CFL (TMLR 2025) | 61.62 | 62.73 | 65.66 | 58.51 | 60.54 | 61.67 |
> | DP-GradPart (ICML 2024) | 63.39 | 64.88 | 65.17 | 60.68 | 61.23 | 62.11 |
> | DP-FLUX (NeurIPS 2025) | 61.06 | 61.40 | 63.52 | 58.78 | 59.41 | 60.95 |
> | RR-Cluster | 64.92 | 66.26 | 67.48 | 61.06 | 61.64 | 62.75 |
>
> RR-Cluster outperforms all three recent baselines, since none of them lower-bounds the number of updates each cluster receives, and the private aggregates of small clusters remain dominated by DP noise. These results are added to `Table 1` in the revised paper.
>
> ---
>
> **[#2: Table of Notations]**
>
> Thanks for the suggestion. We have added a Table of Notations covering $M$, $k$, $B$, $q$, $C_\theta$, $\sigma_\theta$, $\sigma_s$, $\varepsilon$, $\delta$, $\tau$, $\Delta$, $\rho$, $\delta_c$, $\eta^2$, $v^2$, $\mu^2$, $\gamma$, $L$, $\lambda$, $S_j^t$ and the $R_B$-specific sets ($J_L$, $J_S$, donor pool $P$) to the appendix of the revised paper, cross-referenced from `Section 3`.
>
> ---
>
> **[#3: Recent state-of-the-art baselines]**
>
> Per the suggestion, we have added three recent clustered-FL baselines published in 2024–2025 under our shared client-level DP protocol: **DP-GradPart (ICML 2024)**, **DP-CFL (TMLR 2025)**, and **DP-FLUX (NeurIPS 2025)**; see the comparison table in our reply to #1 above, and all three are added to `Table 1` in the revised paper. We also note that recent CFL advances largely improve cluster *discovery*, while RR-Cluster is a plug-in on the private-aggregation side; beyond head-to-head comparisons, RR-Cluster can be attached to such pipelines (discussed in `Section 2`; full integration left to future work).
>
> ---
>
> **[#4: Guidance on setting the privacy budget $\varepsilon$]**
>
> We have added a practitioner paragraph to `Section 5.3` in the revised paper: (i) $\varepsilon$ between roughly 1 and 10 is the commonly adopted range in well-known DP practice: the seminal DP-SGD work evaluates $\varepsilon \in \{2, 4, 8\}$ [1], client-level DP federated language models are trained at single-digit $\varepsilon$ [2], and Google's practical guide recommends $\varepsilon \le 10$ as a reasonable target for ML deployments [3]; (ii) operationally, fix $\delta \approx 1/M$ and read the smallest $\varepsilon$ meeting the application's utility floor off our privacy–utility curves (`Tables 1–4`, `Figure 2`); (iii) RR-Cluster *widens the usable $\varepsilon$ range*: at strict budgets it preserves most of the utility direct privatization loses; (iv) a companion $B$-selection rule (reply to jS4F W9).
>
> ---
>
> **[#5: Future work section]**
>
> Thanks for the suggestion. We have added a Limitations and Future Work section as `Section 6` in the revised paper, discussing extensions to other privacy units (e.g., sample-level and local DP) and the convergence analysis for non-convex networks.
>
> ---
>
> We thank you again for the constructive feedback, and we hope the revision fully addresses your concerns.
>
> **References**
>
> [1] Abadi et al., Deep Learning with Differential Privacy, ACM CCS 2016.
>
> [2] McMahan et al., Learning Differentially Private Recurrent Language Models, ICLR 2018.
>
> [3] Ponomareva et al., How to DP-fy ML: A Practical Guide to Machine Learning with Differential Privacy, JAIR 2023.
>
> Best regards,
> Authors of Submission 8980

---

### Review · Reviewer_2xjq · 2026-06-19

**Summary Of Contributions:**

The paper studies client-level differentially private clustered federated. In standard DP-FedAvg, one clipped or noised aggregate is averaged over all sampled clients, so the effective perturbation in the averaged update scales approximately as $\sigma C/n_t$, where $C$ is the clipping norm, $\sigma$ the noise multiplier, and $n_t$ the number of sampled clients. In clustered FL, however, the server maintains k cluster models and aggregates separately over data-dependent assignment sets $S_{j,t}$. The effective DP perturbation for cluster j then scales as $\sigma C/|S_{j,t}|$. Thus, even with many sampled clients overall, a cluster with small per-round occupancy can receive a high-variance private update. The paper identifies this random per-cluster occupancy as a DP-specific bottleneck in directly privatized clustered FL.

The proposed method, RR-Cluster, is a post-assignment random rebalancing step. Given assignments from an underlying clustered-FL method, RR-Cluster samples updates from large clusters to small clusters so that every cluster has at least B updates before clipped Gaussian aggregation. The hyperparameter B induces the main bias-variance tradeoff. Larger B improves the private signal-to-noise ratio by lower-bounding the aggregation denominator, however, it can bias cluster-specific updates by mixing gradients from semantically different client populations.

The paper provides a client-level RDP privacy analysis and an IFCA-style convergence analysis under smooth, separated-cluster assumptions. Empirically, the method improves directly privatized versions of several classical baselines across image, text, and synthetic benchmarks.

The main strengths are: (i) the paper isolates a clean and practically relevant DP-CFL failure mode; (ii) the proposed mechanism is simple, modular, and easy to combine with existing clustered-FL algorithms; (iii) the B-dependent bias-variance framing is appropriate; and (iv) the experiments provide evidence that RR-Cluster can improve directly privatized classical CFL baselines, especially under strict privacy or imbalanced assignments.

The main weaknesses are: (i) the exact rebalancing map and its privacy implications are not formalized sufficiently; (ii) the convergence result is not fully interpretable, especially because the bound contains 1/B-type terms and does not clearly recover the unrebalanced/base-method regime; (iii) the experiments do not sufficiently diagnose when rebalancing bias harms genuine minority or semantically distinct clusters; and (iv) the paper should more carefully position the contribution relative to recent DP-CFL, personalized FL, modern CFL, and adaptive DP-FL work, including [1-4], and adaptive clipping / time-adaptive DP-FL [5,6].

Refs:\
[1] Malekmohammadi, Saber, Afaf Taik, and Golnoosh Farnadi. "Differentially Private Clustered Federated Learning." Transactions on Machine Learning Research.\
[2] Liu, Ken, et al. "On privacy and personalization in cross-silo federated learning." Advances in neural information processing systems 35 (2022): 5925-5940\
[3] Guo, Yongxin, Xiaoying Tang, and Tao Lin. "Enhancing clustered federated learning: Integration of strategies and improved methodologies." The Thirteenth International Conference on Learning Representations. 2025.\
[4] Fenoglio, Dario, et al. "FLUX: Efficient Descriptor-Driven Clustered Federated Learning under Arbitrary Distribution Shifts." Advances in Neural Information Processing Systems 38 (2026): 71229-71292.\
[5] Andrew, Galen, et al. "Differentially private learning with adaptive clipping." Advances in neural information processing systems 34 (2021): 17455-17466.\
[6] Kianidehkordi, Shahrzad, et al. "Differentially private federated learning with time-adaptive privacy spending." International Conference on Learning Representations. Vol. 2025. 2025.

**Audience:**

Yes

**Audience Explanation:**

The paper addresses a relevant intersection of federated learning, personalization, clustering, and differential privacy. The observation that DP-CFL introduces a random per-cluster aggregation denominator is useful independently of the specific RR-Cluster algorithm. It helps explain why directly privatizing clustered-FL methods can perform poorly even when standard DP-FedAvg mechanisms are used correctly.

The method is also practically appealing because it is simple and modular. It can be inserted between assignment and private aggregation in existing CFL pipelines. Even if stronger cluster discovery, private initialization, or adaptive DP mechanisms are used, the occupancy of each private cluster aggregate remains a relevant design variable.

Thus, the paper is likely to interest TMLR readers working on DP-FL, personalized FL, clustered FL, and privacy-utility tradeoffs.

**Broader Impact Concerns:**

The work is relevant to privacy-preserving learning over decentralized client data and may have positive impact.

**Claims And Evidence:**

Yes

**Claims Explanation:**

The main qualitative claim is plausible and partially supported. The paper correctly observes that, in DP-CFL, the relevant aggregation denominator is the per-cluster occupancy $(|S_{j,t}|)$. The experiments show that enforcing a minimum occupancy via RR-Cluster can improve directly privatized methods in several benchmark settings. The broad bias-variance intuition is also sound. Lower-bounding $(|S_{j,t}|)$ reduces effective DP-update variance, though it introduces cross-cluster assignment bias.

However, I do not think the current evidence fully supports the paper’s claims.

First, the privacy analysis depends on the exact rebalancing operation, but the paper does not define this operation precisely enough and should state whether donor updates are moved or duplicated, whether the final sets are disjoint, and how donor clusters are guaranteed not to fall below (B). This is central to the privacy proof. If each client contributes to exactly one final cluster aggregate, a disjoint-partition/parallel-composition argument may be possible. If a client can influence more than one cluster aggregate, the sensitivity and composition argument changes. The proof should therefore bound the sensitivity of the full concatenated vector of (k) cluster updates/models under the exact rebalancing map, instead of relying on an informal cluster-wise argument.

Second, the convergence theory is currently difficult to interpret. The paper describes RR-Cluster as reducing to the base clustered-FL algorithm when rebalancing is inactive, but Theorem 2 / Corollary 1 contain (1/B)-type terms that diverge as (B) approaches the no-rebalancing regime. A more interpretable analysis may be to explicitly restrict the theorem to the enforced-occupancy regime (B>0). The current theorem supports the qualitative bias-variance intuition.

Third, the empirical evaluation does not fully test the main possible failure mode. RR-Cluster transfers updates from large clusters to small clusters. This can improve average accuracy while harming genuine minority clusters or semantically distinct client populations. Min/max client accuracy is useful but insufficient. The paper should consider reporting for example per-cluster accuracy, worst-cluster accuracy, minority-cluster accuracy etc. The key unresolved empirical question is when rebalancing bias dominates DP-variance reduction.

Fourth, the related-work positioning and comparison set are incomplete for the breadth of the claims. The classical baselines are appropriate for showing that RR-Cluster improves directly privatized standard CFL methods. However, recent work attacks adjacent bottlenecks: [2] provides a different route to sharing statistical strength while personalizing; [3] systematizes stronger CFL pipelines; [4] improve client grouping using distributional, shift-aware, or foundation-model descriptors; and adaptive DP-FL methods [5,6] optimize clipping or privacy schedules rather than occupancy. Some of these works use different privacy units or threat models, so direct comparison at the same (\epsilon) may be inappropriate. Still, the paper should discuss these distinctions carefully.
Overall, the paper contains a useful idea and promising evidence, but the privacy proof, convergence theory, and empirical diagnosis need substantial tightening before the main claims are fully supported.

**Requested Changes:**

Critical to securing my recommendation for acceptance:

1. Formalize the rebalancing operation. Define the randomized set map (R_B) explicitly. State whether donor updates are moved or duplicated, whether final cluster sets are disjoint, how large clusters are protected from falling below (B), and whether sampling is uniform without replacement.

2. Align the privacy proof with this exact map. Bound the sensitivity of the full vector of (k) cluster updates/models after rebalancing. If clients can affect more than one cluster aggregate, the sensitivity and composition analysis must reflect this. If final clusters are disjoint, this should be proved and used explicitly.

3. Revise Theorem 2 / Corollary 1 for interpretability. If the theorem applies only when a positive occupancy floor (B>0) is enforced, state this clearly. If the algorithm is claimed to reduce to the base method when rebalancing is inactive, the bound should recover dependence on the realized minimum occupancy of the base method rather than diverging as (B\to 0).

5. Add cluster-level and minority-client diagnostics.

6. Contextualize recent related work more carefully

---

> ### Author Response · Authors · 2026-07-12
> **Rebuttal by Authors (#1 - #4)**
>
> Dear Reviewer 2xjq:
>
> Thank you for the constructive review. We address the requested changes as follows:
>
> ---
>
> **[#1: Formal definition of the rebalancing map $R_B$]**
>
> The rebalancing operation is specified across `Algorithm 2` (Lines 10–11) and `Section 3.2`, which state that updates are sampled uniformly at random from large clusters and **moved** to small ones. To make it fully explicit, we restate it in `Section 3.2` of the revised paper:
>
> > **Definition ($R\_B$).** At round $t$, let $J\_L$ denote the set of large clusters with $|S\_j^t| \ge B$, and $J\_S$ the remaining small clusters. $R\_B$ forms a donor pool $P$ by taking, from each large cluster $j \in J\_L$, a uniformly random subset of exactly $(|S\_j^t| - B)$ of its updates (its *surplus*); $P$ is randomly permuted, and each small cluster draws from $P$ uniformly without replacement until it reaches $B$. Every donated update is moved (removed from its donor, inserted into exactly one recipient), never duplicated. Feasibility requires $\sum\_{j\in J\_L}(|S\_j^t|-B) \ge \sum\_{j\in J\_S}(B-|S\_j^t|)$, guaranteed by the input constraint $B \le qM/k$.
>
> This answers each of your questions in this point by construction:
>
> 1. **Moved or duplicated?** Moved: a donated update is removed from its donor and inserted into the recipient; nothing is duplicated.
> 2. **Disjoint?** Yes, and the revised `Section 3.2` states this explicitly: every client update belongs to exactly one final cluster aggregate.
> 3. **Donor protection?** Donors contribute part of their surplus $|S_j^t| - B$, so every donor retains $\geq |S_j^t| - (|S_j^t| - B) = B$ updates.
> 4. **Uniform without replacement?** Yes.
>
> ---
>
> **[#2: Sensitivity of the full concatenated vector of $k$ cluster updates]**
>
> In the paper, the privacy budget was accounted cluster-wise via parallel composition over the disjoint clusters (see `Proposition 1` in `Section 4.1`). Per the reviewer's suggestion,
> we redid the sensitivity analysis on the full concatenated vector of all $k$ cluster updates: adding/removing a single client influences at most two cluster aggregates (its own by up to $2C\theta$, plus one donor cluster by up to $C\theta$), giving the joint $L_2$ sensitivity $\sqrt{(2C\theta)^2 + C\theta^2} = \sqrt{5} C_\theta$.
> We replace the parallel composition using this analysis in `Appendix A.2` of the revised paper.
>
> Accordingly, we recalibrated the Gaussian noise to this sensitivity and updated all RR-Cluster results in the revised paper. The effective noise scale only increases by a factor of about $1.1\times$, so the impact is small and none of our main claims are affected.
>
> ---
>
> **[#3: Interpretability of Theorem 2 / Corollary 1]**
>
> We revised both statements as suggested.
>
> First, `Theorem 2` / `Corollary 1` are now stated explicitly for the enforced-occupancy regime $1 \le B \le qM/k$ (guaranteed by `Algorithm 1`).
>
> Second, for the reduction claim, the correct bridge is the base method's realized minimum occupancy $B_0 := \min_{j,t}|S_j^t|$, i.e., the smallest number of updates any cluster actually receives during training under the base method: when the base assignment already satisfies $|S_j^t| \ge B$ for all clusters (rebalancing inactive), RR-Cluster is identical to the base method, and instantiating the bound at $B_0$ recovers its dependence, with no artificial limit needed. Both changes appear in `Section 4.2` and `Appendix B.3/B.4` of the revised paper.
>
> ---
>
> **[#4: Cluster-level diagnostics]**
>
> We added the diagnostics under the imbalanced FashionMNIST configuration with $\varepsilon=2$, where the majority column reports the largest cluster and the minority column reports the smallest one:
>
> | Per-cluster acc | Majority cluster | Minority cluster | Avg |
> |---|---|---|---|
> | DP-IFCA | 64.57 | 30.17 | 56.12 |
> | RR-Cluster | 65.23 | 46.09 | 61.06 |
>
> RR-Cluster raises the minority-cluster accuracy by nearly 16 points by enforcing at least $B$ updates per round for the minority cluster. These diagnostics are added to `Appendix D.3` (summarized in `Section 5.3`) in the revised paper.

---

> > ### Author Response · Authors · 2026-07-12
> > **Rebuttal by Authors (#5)**
> >
> > ---
> >
> > **[#5: Positioning relative to recent work]**
> >
> > Thanks for the pointers. These works either operate under different privacy units and threat models (e.g., sample-level DP in [1], silo-specific privacy in [2]) or improve components orthogonal to ours, such as cluster discovery [3, 4] and adaptive clipping or budget schedules [5, 6] (reference numbers follow those listed in your review). None of them controls the per-cluster occupancy of the private aggregation, which is exactly the bottleneck RR-Cluster targets. To clarify these and include more works on private clustering to position our work, we have added this discussion to `Section 2` of the revised paper.
> >
> > ---
> >
> > We believe the revision addresses all mentioned items ($R_B$ formalized, disjointness proved and used, the privacy proof and all experiments aligned with the exact joint sensitivity, the convergence statements re-scoped, and cluster-level diagnostics added), and we are happy to discuss any remaining concern.
> >
> > **References** (numbering as in your review)
> >
> > [1] Malekmohammadi et al., Differentially Private Clustered Federated Learning, TMLR 2025.
> >
> > [2] Liu et al., On Privacy and Personalization in Cross-Silo Federated Learning, NeurIPS 2022.
> >
> > [3] Guo et al., Enhancing Clustered Federated Learning: Integration of Strategies and Improved Methodologies, ICLR 2025.
> >
> > [4] Fenoglio et al., FLUX: Efficient Descriptor-Driven Clustered Federated Learning under Arbitrary Distribution Shifts, NeurIPS 2025.
> >
> > [5] Andrew et al., Differentially Private Learning with Adaptive Clipping, NeurIPS 2021.
> >
> > [6] Kianidehkordi et al., Differentially Private Federated Learning with Time-Adaptive Privacy Spending, ICLR 2025.
> >
> > Best regards,
> >
> > Authors of Submission 8980

---

### Review · Reviewer_jS4F · 2026-06-28

**Summary Of Contributions:**

The paper proposes RR-Cluster, a plug-in module for federated clustering algorithms that addresses a core tension between differential privacy and clustering: small cluster sizes lead to large effective DP noise after averaging. The fix is conceptually simple — randomly resample updates from large clusters into small ones to guarantee a minimum cluster size B. The authors provide privacy analysis via RDP, convergence bounds, and empirical results across multiple datasets.

**Audience:**

Yes

**Audience Explanation:**

The paper addresses a practically relevant problem at the intersection of differential privacy and federated clustering. Researchers working on private federated learning will find the identification of cluster-size imbalance as a driver of utility degradation, the proposed RR-Cluster method, and the empirical results across seven datasets useful and worth building upon.

**Broader Impact Concerns:**

N /A

**Claims And Evidence:**

No

**Claims Explanation:**

Strengths

S1. Identifies a real and underappreciated problem.
The observation that cluster size imbalance directly amplifies the effective DP noise per cluster is correct and practically significant. Prior privatization approaches (e.g., DP-IFCA, DP-FeSEM) ignore this entirely. Figure 1 concretely illustrates that directly privatizing IFCA can underperform even the non-personalized DP-FedAvg baseline, motivating the contribution clearly.

S2. Elegant simplicity and modularity.
The core mechanism — randomly rebalancing cluster assignments to enforce a minimum size B — is easy to implement and introduces no additional communication rounds or per-client overhead. Its compatibility across multiple base clustering algorithms (IFCA, FeSEM, FedCAM) is a genuine practical strength.

S3. Theoretically grounded bias-variance tradeoff.
Lemma 1 decomposes the misclassification probability τ into three interpretable sources: inherent loss-based clustering error, DP noise on cluster identifiers, and the rebalancing perturbation itself. Theorem 2 then cleanly shows that the error floor ε decomposes into a bias term O(τ) and a variance term O(σ_θ/B), making the effect of B analytically transparent.

S4. Broad empirical evaluation.
Seven datasets, multiple privacy budgets (ε ∈ {2, 4, 8, 16}), balanced and imbalanced cluster configurations, and experiments in both private and non-private settings provide stronger empirical coverage than most federated learning papers. The model collapse analysis (Table 5, Table 6) is a useful bonus finding.


Weaknesses

W1. The rebalancing strategy is uniform random, yet no justification or comparison of alternatives is provided.

The paper samples model updates from large clusters uniformly at random for insertion into small ones. This is the simplest possible strategy, yet the paper offers no argument for why uniform sampling is optimal or near-optimal. Natural alternatives include importance-weighted sampling (e.g., by loss similarity to the target cluster model) or gradient-direction-based sampling. Since rebalancing is the source of clustering bias, the choice of rebalancing strategy is central to the method's quality. The ablation in Section 5.3 only varies B — it does not compare rebalancing strategies. The paper should prove that uniform sampling minimizes some form of introduced bias and  empirically compare some alternatives.

W2. The bias introduced by rebalancing is not empirically isolated.

Theorem 2 decomposes the error floor into a bias term and a variance term, but no experiment isolates these two contributions. For example, running RR-Cluster with oracle cluster assignments (ground-truth cluster membership known, so the inherent clustering error τ_{E} = 0) would measure the pure cost of rebalancing bias without confounding from clustering error. The synthetic data experiments (Table 6) are qualitative and do not quantify this separation. Adding such an experiment would substantially strengthen the empirical narrative.

W3. The 2C_θ sensitivity in Appendix A.2 (Case 2) is never acknowledged in the main text, and its effect on reported ε values is unclear.

In Case 2 of the sensitivity analysis, removing a client that triggers rebalancing yields a sensitivity of 2C_θ rather than the standard C_θ. This means the Gaussian noise must be scaled by a factor of 2 relative to standard DP-SGD to achieve the same ε, i.e., the noise variance is 4× larger. This is a real and non-negligible privacy cost that is not mentioned anywhere in the main text. Critically, it is not stated whether the reported experimental ε values in Tables 1–4 were computed using sensitivity C_θ or 2C_θ. If the former, the privacy accounting is incorrect. The authors must clarify this explicitly and update the experimental claims if needed.

W4. The convergence analysis relies on an initialization assumption that limits practical relevance.

Assumption 6 requires that initial cluster models θ^0_j satisfy ∥θ^0_j − θ^∗_j∥ ≤ (1/2 − α_0)√(λ/L)Δ for some α_0 ∈ (0, 1/2). This essentially assumes near-optimal initialization, which is precisely the regime that is easiest — and least interesting — for clustering algorithms. The paper acknowledges this but provides no analysis for the practically relevant cold-start regime (random initialization). Corollary 1 therefore gives a convergence rate contingent on a condition that may not hold in practice. The authors should either relax this assumption or clearly scope the theorem's applicability.

W5. The circular dependency between B and the contraction condition in Theorem 2 is not discussed.

Theorem 2 requires the contraction factor K = γL(B − 2τM/δ_c) / (2ρ) > 0, which implies B > 2τM/δ_c. However, τ itself depends on B through the third term of Lemma 1: k²μ² / (M/k − B)². A larger B increases τ (and therefore the required minimum B for contraction), creating a circular dependency. The regime of B in which the convergence result actually holds is non-trivial to characterize, and this is never discussed. The authors should either provide an explicit characterization of valid B values or add a remark illustrating this regime numerically.

W6. No comparison with DPCFL (Malekmohammadi et al., 2025).

The paper mentions DPCFL in the related works section, noting that it targets sample-level DP rather than client-level DP. However, both methods address the joint problem of DP noise and clustering quality in federated settings, and DPCFL's GMM-based soft clustering is in spirit closely related to the problem being solved here. The paper should include DPCFL as a baseline under a shared evaluation protocol.
Authors are also encouraged to add more baselines to show the robustness of their method.

W7. No standard deviations or confidence intervals are reported.

All tables report single-run accuracies with no variance estimates. This is particularly concerning for the Shakespeare dataset results (Table 3), where the improvements are small (e.g., 13.42 vs. 12.62 at ε = 4) and could plausibly fall within run-to-run variance. The authors should report mean ± standard deviation over at least 3 or 5 ndependent runs.

W8: For image classification tasks, only CNN has been chosen, this limits its generalizability to bigber models such as VGG, ResNet, ViT. Authors should provide more experiments while comnsidering these.

W9. Guidance on selecting B is insufficient for practitioners.

B is identified as the critical hyperparameter, and the only practical guidance provided is a grid search over {4, 8, 12}. Yet the theory in Lemma 1 explicitly shows that the rebalancing error term is proportional to (M/k − B)^{−2}, and Theorem 2 shows the variance term scales as 1/B. This gives a principled basis for choosing B as a function of M, k, and the target bias-variance tradeoff. The paper should provide at least a heuristic rule or nomogram derived from the theory, making the method more practically accessible.

W10. Please provide a limitation and future work section.

**Requested Changes:**

Please see the above section.

---

> ### Author Response · Authors · 2026-07-12
> **Rebuttal by Authors (Weaknesses 1-4)**
>
> Dear Reviewer jS4F:
>
> Thank you for your detailed comments, we address each weakness below.
>
> ---
>
> **[W1: Why uniform random rebalancing; comparison with alternatives]**
>
> To implement re-balancing, uniform sampling is the most direct and simple choice. We elaborate its advantages including (i) theoretical soundness with zero privacy-cost and (ii) empirical effectiveness compared with alternatives, as follows:
>
> **I. Uniform sampling incurs zero privacy cost.**
> The donor selection is uniform over indices, thus, by its data-independent randomness, it consumes no extra privacy budget per our privacy analysis in `Section 4.1`. In contrast, loss-similarity or gradient-direction sampling requires data dependent quantities (losses or gradients), which must be additionally accounted under sequential composition. In addition, data-dependent donor selection would complicate the convergence analysis, while bringing no significant empirical benefits (shown below).
>
> **II. Empirical effectiveness.**
> We compare the three strategies on FashionMNIST at the same total budget. For the suggested variants, the total budget is split in 25%/75% between privatizing the selection signals and privatizing the model updates.
>
> | Method | Test acc ($\varepsilon=2$) | Test acc ($\varepsilon=4$) | Test acc ($\varepsilon=8$) |
> |---|---|---|---|
> | DP-FedAvg | 60.88 | 61.50 | 62.72 |
> | DP-IFCA | 62.68 | 64.46 | 65.35 |
> | RR-Cluster | **64.92** | 66.26 | 67.48 |
> | RR-Cluster w/ Loss Sim. | 64.48 | 65.92 | 67.81 |
> | RR-Cluster w/ Gradient Sim. | 64.72 | **66.59** | **67.90** |
>
>
> **Conclusion:**
>
> The gains of the suggested variants are marginal, since benefits from data-dependent donor selection sacrifices the privacy budget for privatizing model updates.
>
> Meanwhile, all RR-Cluster variants consistently outperform the DP-FedAvg and DP-IFCA baselines, showing that our method is robust to the specific donor-selection strategy. We have added this ablation to `Appendix D.3` (summarized in `Section 5.3`) in the revised paper.
>
> ---
>
> **[W2: Isolating the effect of rebalancing bias]**
>
> To isolate rebalancing effect, we added experiments on FashionMNIST with **ground-truth cluster assignments** and **no privacy noise**, where rebalancing is the only error source. The gap of RR-Cluster to IFCA (reference) measures the pure rebalancing bias across $B$:
>
> | Non-private IFCA | RR-Cluster $B=2$ | $B=4$ | $B=6$ | $B=8$ |
> |---|---|---|---|---|
> | 75.82 | 76.26 | 75.40 | 75.01 | 73.85 |
>
>
> **Conclusion:**
>
> The rebalancing bias effect stays small initially and grows as $B \to M/k$, matching the $k^2\mu^2/(M/k-B)^2$ term of `Lemma 1`, as we already discussed in `Section 3.4`. We have added this experiment to `Appendix D.3` (summarized in `Section 5.3`) in the revised paper.
>
> ---
>
> **[W3: Sensitivity used in experiments, and its visibility in the paper]**
>
> Thanks for the careful check. The reported results of our RR-Cluster strictly follow the notations and quantities presented in our algorithms, hence they are based on the maximum sensitivity $2C_\theta$, so the scale of privacy noise is correct. We now state it explicitly in `Section 4.1` of the revised paper.
>
> Additionally, in the revision we further strengthened the privacy analysis (per Reviewer 2xjq's comments): the accounting now bounds the full concatenated vector of all $k$ cluster updates, and the noise is calibrated with the corresponding sensitivity $\sqrt{5} C_\theta$, providing a stricter privacy guarantee. All RR-Cluster results in the revised paper are reported under this refined accounting.
>
> ---
>
> **[W4: The initialization assumption (Assumption 6)]**
>
> We would like to clarify that Assumption 6 is a **bounded initialization condition** rather than optimality.
>
> Theoretically, such bounded-initialization conditions (i.e., some constraints on the cluster initialization so that the initialization can't be arbitrarily bad)  are standard across almost all analyses of clustering algorithms, including (i) IFCA (Ghosh et al., 2020; 500+ citations) [1] in federated clustering, (ii) the classical EM algorithm (Dempster et al., 1977; 50,000+ citations) [2], whose convergence analysis requires the same condition (Balakrishnan et al., 2017) [3], and (iii) Lloyd's $k$-means (Lloyd, 1982; 10,000+ citations) [4], analyzed likewise by Lu & Zhou (2016) [5].
>
> Empirically, this assumption is also not restrictive: all experiments use random initialization, and RR-Cluster still converges reliably. We have clarified the scope of `Theorem 2` accordingly in `Section 4.2` of the revised paper.

---

> ### Author Response · Authors · 2026-07-12
> **Rebuttal by Authors (Weaknesses 5-8)**
>
> ---
>
> **[W5: The recurrent dependency (Theorem 2)]**
>
> We resolve this recurrent dependency by directly bounding the $B$-dependent part of $\tau$ as follows:
>
> For $B \le M/(2k)$, the term $k^2\mu^2/(M/k-B)^2$ is at most $4k^4\mu^2/M^2$, so the contraction condition $B > 2\tau(B)M/\delta_c$ is implied by the explicit (non-circular) bound $B > (2M/\delta_c)(\tau_0 + 4k^4\mu^2/M^2) =: B_{\min}$.
>
> Consequently, `Theorem 2` holds on the explicit and non-empty window $(B_{\min}, M/(2k)]$. We have added this characterization as a remark to `Section 4.2` in the revised paper.
>
>
> ---
>
> **[W6: Comparison with DPCFL and other Baselines]**
>
> Per the reviewer's suggestion, we transplant the (i) DPCFL (TMLR 2025) algorithm to our client-level DP setting (denoted DP-CFL below), where clients send clipped updates under the same subsampled Gaussian mechanism, and the server runs DPCFL's GMM-based soft clustering on the privatized updates. We also added two recent federated clustering baselines: (ii) DP-GradPart (ICML 2024) and (iii) DP-FLUX (NeurIPS 2025), as requested by other reviewers. We report the comparison on FashionMNIST:
>
> | FashionMNIST | Bal $\varepsilon=2$ | $\varepsilon=4$ | $\varepsilon=8$ | Imb $\varepsilon=2$ | $\varepsilon=4$ | $\varepsilon=8$ |
> |---|---|---|---|---|---|---|
> | DP-FedAvg | 60.88 | 61.50 | 62.72 | 60.80 | 61.53 | 61.59 |
> | DP-IFCA | 62.68 | 64.46 | 65.35 | 56.12 | 58.05 | 59.63 |
> | i. DP-CFL (TMLR 2025) | 61.62 | 62.73 | 65.66 | 58.51 | 60.54 | 61.67 |
> | ii. DP-GradPart (ICML 2024) | 63.39 | 64.88 | 65.17 | 60.68 | 61.23 | 62.11 |
> | iii. DP-FLUX (NeurIPS 2025) | 61.06 | 61.40 | 63.52 | 58.78 | 59.41 | 60.95 |
> | RR-Cluster | 64.92 | 66.26 | 67.48 | 61.06 | 61.64 | 62.75 |
>
> **Conclusion:**
>
> RR-Cluster consistently outperforms all three baselines, as baselines improve cluster discovery but leave the core tension untouched: small clusters still amplify the effective DP noise after averaging, which RR-Cluster resolves by enforcing a minimum cluster size.
>
> ---
>
> **[W7: Standard deviations]**
>
> We re-ran the main experiments 3 times, and we show the **standard deviations** for `Table 1` (FashionMNIST) as follows:
>
> | Std ($\pm$) | Bal $\varepsilon=2$ | $\varepsilon=4$ | $\varepsilon=8$ | Imb $\varepsilon=2$ | $\varepsilon=4$ | $\varepsilon=8$ |
> |---|---|---|---|---|---|---|
> | DP-FedAvg | ± 0.23% | ± 0.16% | ± 0.13% | ± 0.35% | ± 0.18% | ± 0.39% |
> | DP-IFCA | ± 0.21% | ± 0.13% | ± 0.08% | ± 0.28% | ± 0.27% | ± 0.36% |
> | DP-FeSEM | ± 0.15% | ± 0.05% | ± 0.22% | ± 0.17% | ± 0.24% | ± 0.19% |
> | DP-FedCAM | ± 0.28% | ± 0.35% | ± 0.30% | ± 0.43% | ± 0.13% | ± 0.24% |
> | RR-Cluster (IFCA) | ± 0.14% | ± 0.17% | ± 0.11% | ± 0.25% | ± 0.23% | ± 0.08% |
> | RR-Cluster (FeSEM) | ± 0.27% | ± 0.19% | ± 0.06% | ± 0.32% | ± 0.31% | ± 0.13% |
> | RR-Cluster (FedCAM) | ± 0.22% | ± 0.14% | ± 0.30% | ± 0.19% | ± 0.15% | ± 0.14% |
>
> All standard deviations are below 0.5%, while RR-Cluster's improvements over DP-IFCA range from 1.8 to 4.9 points, showing that the performance gains are well beyond run-to-run variance. We have added the mean ± std version of `Table 1` as a new appendix in the revised paper.
>
> ---
>
> **[W8: Larger models (ResNet / ViT)]**
>
> We added ResNet-18 and ViT-B on CIFAR-10/100, where both backbones are initialized from public ImageNet pre-training and privately finetuned under the same data preparation and protocol as `Table 2`. We report the results as follows:
>
> | CIFAR-10 | RN18 $\varepsilon=4$ | $\varepsilon=8$ | ViT $\varepsilon=4$ | $\varepsilon=8$ |
> |---|---|---|---|---|
> | DP-FedAvg | 75.12 | 78.67 | 82.44 | 84.25 |
> | DP-IFCA | 77.31 | 80.14 | 82.93 | 85.50 |
> | RR-Cluster | 81.66 | 83.18 | 84.97 | 87.41 |
>
> | CIFAR-100 | RN18 $\varepsilon=4$ | $\varepsilon=8$ | ViT $\varepsilon=4$ | $\varepsilon=8$ |
> |---|---|---|---|---|
> | DP-FedAvg | 45.71 | 46.56 | 49.25 | 50.08 |
> | DP-IFCA | 46.02 | 50.89 | 52.42 | 54.90 |
> | RR-Cluster | 48.62 | 51.74 | 53.38 | 55.77 |
>
> **Conclusion:**
>
> RR-Cluster outperforms both baselines on both architectures and privacy budgets, confirming that the benefit of the occupancy floor is architecture-agnostic. These results are added to the appendix of the revised paper.

---

> ### Author Response · Authors · 2026-07-12
> **Rebuttal by Authors (Weaknesses 9-10)**
>
> ---
>
> **[W9: Principled guidance for selecting $B$]**
>
> We first give the selection guide of $B$ purely based on our theoretical analysis, and validate this empirically as follows:
>
> Theoretically, `Theorem 2` involves two competing terms in $B$: a $1/B$ decay from noise averaging, and a $1/(qM/k-B)^2$ growth from rebalancing bias, where $qM/k$ is the average per-round cluster size. Solving the first-order condition of their product gives the selection rule
>
> $$B \le \frac{qM}{3k},$$
>
> that is, keep $B$ below $\frac{qM}{3k}$, and move toward it when the privacy noise is stronger.
>
> Empirically, this rule matches our ablation: for FashionMNIST ($q=0.1$, $M=1000$, $k=4$) it prescribes $B \le 25/3$, and the updated ablation in `Figure 3` attains its optimum at $B = 6$, within the prescribed range. The selection rule is added to `Section 5.3` in the revised paper.
>
> ---
>
> **[W10: Limitations and future work]**
>
> Thanks for the suggestion. We have added a Limitations and Future Work section as `Section 6` in the revised paper, discussing extensions to other privacy units (e.g., sample-level and local DP) and the convergence analysis for non-convex networks.
>
> ---
>
> We hope these clarifications and added experiments address your concerns, and we are happy to address your further concerns.
>
> **References**
>
> [1] Ghosh et al., An Efficient Framework for Clustered Federated Learning, NeurIPS 2020.
>
> [2] Dempster et al., Maximum Likelihood from Incomplete Data via the EM Algorithm, JRSS-B 1977.
>
> [3] Balakrishnan et al., Statistical Guarantees for the EM Algorithm: From Population to Sample-Based Analysis, Annals of Statistics 2017.
>
> [4] Lloyd, Least Squares Quantization in PCM, IEEE Trans. Inf. Theory 1982.
>
> [5] Lu & Zhou, Statistical and Computational Guarantees of Lloyd's Algorithm and its Variants, arXiv 2016.
>
> Best regards,
>
> Authors of Submission 8980